

# Measurements of Global Distributions of Polar Mesospheric Clouds during 2005-2012 by MIPAS/Envisat

Maya García-Comas[1], Manuel López-Puertas[1], Bernd Funke[1], Á. Aythami Jurado-Navarro[1], Angela Gardini[1], Gabriele P. Stiller[2], Thomas v. Clarmann[2], and Michael Höpfner[2]

[1]Instituto de Astrofísica de Andalucía, CSIC, Granada, Spain
[2]Karlsruhe Institute of Technology, Institute for Meteorology and Climate Research, Karlsruhe, Germany

*Correspondence to:* M. López-Puertas (puertas@iaa.es)

**Abstract.** We have analysed the MIPAS IR measurements of PMCs for the summer seasons in the Northern and Southern Hemispheres from 2005 to 2012. Measurements of PMCs using this technique are very useful because they are sensitive to the total ice volume independent of particle size. For the first time, MIPAS has provided coverage of the PMCs total ice volume from mid-latitudes to the poles. MIPAS measurements indicate the existence of a continuous layer of mesospheric
ice, extending from about ∼81 km up to about 88-89 km on average and from the poles to about 50–60° in each hemisphere, increasing in concentration with proximity to the poles. We have found that the ice concentration is larger in the Northern Hemisphere than in the Southern Hemisphere. The ratio between the ice water content (IWC) in both hemispheres is also latitude-dependent, varying from a NH/SH ratio of 1.4 close to the poles to a factor of 2.1 around 60°. This also implies that PMCs extend to lower latitudes in the NH. A very clear feature of the MIPAS observations is that PMCs tend to be at higher
altitudes with increasing distance from the polar region (in both hemispheres), particularly equator-wards of 70°, and that they are about 1 km higher in the SH than in the NH. The difference between the mean altitude of the PMC layer and the mesopause altitude increases towards the poles and is larger in the NH than in the SH. The PMC layers are denser and wider when the frost point temperature occurs at lower altitudes. The layered water vapour structure caused by sequestration and by sublimation of PMCs is more pronounced at latitudes northernmost of 70 degrees. Finally, MIPAS observations have also shown a clear
impact of the migrating diurnal tide on the diurnal variation of the PMCs ice concentration.

## 1 Introduction

Polar mesospheric clouds (PMCs), usually called noctilucent clouds (NLCs) when observed from the ground, occur at the coldest regions of the atmosphere near the summer high latitude mesopause. PMCs normally form a layer extending vertically for several kilometres, peaking near 83 km, located at latitudes poleward of 50°. In this region the temperature frequently drops
below the frost point which, for mesospheric pressures and humidities, is as low as 150 K. They mainly consist of water ice particles with radii ranging from a few nm to about 100 nm (Rusch et al., 1991; Gumbel and Witt, 1998; Hervig et al., 2001; von Savigny et al., 2005).



NLCs are only the optically visible (and lower) part of the layer of icy particles covering the entire polar mesopause region (Berger and Zahn, 2002; Rapp and Thomas, 2006). However, the whole layer modifies the ambient plasma of the D-region and gives rise to intense radar echoes, the so-called PMSE (Polar Mesospheric Summer Echoes) (Rapp and Lübken, 2004). It is now generally accepted that the larger particles are located near the bottom of the layer, while the smaller ones are more likely

to be near the top of the layer (Berger and Zahn, 2002; von Savigny et al., 2005; Baumgarten and Fiedler, 2008).

PMCs have been intensively studied using ground, rocket, and space observations (SNOE, SBUV, ODIN, SCIA-MACHY, GOMOS, AIM) (Baumgarten and Fiedler, 2008; Fiedler et al., 2009; Gumbel and Witt, 1998; Bailey et al., 2005; DeLand et al., 2003; Petelina et al., 2006; von Savigny et al., 2005, 2007; von Savigny and Burrows, 2007; Pérot et al., 2010; Russell III et al., 2009); as well as sophisticated models (Berger and Zahn, 2002; Berger and von Zahn, 2007). A good review

on our knowledge about PMCs up until 2006 was compiled by Rapp and Thomas (2006). A more recent review, including a comparison with mesospheric clouds on Mars, was conducted by Määttänen et al. (2013).

PMCs are being discussed as potential early indicators of global climate change (Thomas et al., 1989; von Zahn, 2003) because they are very sensitive to temperature and water vapour concentration. Since enhanced $CO_2$ amounts (see, e.g., Yue et al., 2015) would lead to an eventual cooler upper mesosphere/lower thermosphere, and higher $CH_4$ amounts may lead to enhanced

$H_2O$ near the mesopause (Roble and Dickinson, 1989; Nedoluha et al., 2009; Garcia et al., 2015), they could both lead to an increase of PMC occurrence, which might be interpreted as an effect of climate change in the upper atmosphere. There is not, however, a consensus in the scientific community about this aspect (von Zahn, 2003; Thomas, 2003). The recent study of SBUV (Solar Backscatter Ultraviolet) data from 1979 through 2013 by DeLand and Thomas (2015) has shown, in addition to the clear solar cycle signal, a good correlation with stratospheric ozone variations. Also, they have found that PMC ice water

content in bright clouds increased rapidly from 1979 through the late 1990s and has been approximately constant from the late 1990s through 2013.

Hervig and Stevens (2014) calculated SBUV ice water content (IWC) values using a different method to DeLand and Thomas (2015) and compared their results with SOFIE data. They found good agreement in average IWC if an appropriate threshold was applied to the SOFIE data set, and consistent day-to-day and year-to-year variations between

both data sets.

Russell et al. (2014) looked at trends in the northern mid-latitude noctilucent cloud occurrences using satellite data and model simulations and found a significant increase in the PMC occurrences at mid-latitudes from 2002 to 2011. This result differs somewhat from the insignificant trend found by DeLand and Thomas (2015) for a similar period but at higher latitudes.

Berger and Lübken (2015) analysed trends in mesospheric ice layers in the high latitude Northern Hemisphere for the 1961–

2013 period with model simulations. They reported a generally good agreement between long-term PMC variations from the MIMAS model and the SBUV satellite observations. They found that the modelled trends in ice water content are latitudinally dependent with no clear trend at mid-latitudes (50°-60°N) but with a clear positive trend at high latitudes (74°-82°N) and also in extreme PMC events.

Thomas et al. (2015) have studied the solar-induced 27-day variations in polar mesospheric clouds using 15 seasons of data

taken by the SOFIE instrument and suggested that the changes in the PMCs are due to 27-day variations of vertical winds.



As described above, a large fraction of the observations taken so far were performed by measuring the scattered light, in the visible or UV, of the solar radiation (in the case of instruments from space) or of the lidar light (in case of ground instruments). This technique usually observes the ice particles with radii larger than about 20 nm but lacks sensitivity for smaller particles (see, e.g., Rapp and Thomas, 2006). A different technique, however, has been used recently by the AIM/SOFIE (Aeronomy of

Ice in the Mesosphere/Solar Occultation for Ice Experiment) instrument. These measurements have provided key characteristics of PMC's such as their frequency, mass density, particle shape, and size distribution, as well as their seasonal evolution and altitude dependence (see, e.g. Hervig et al., 2009a, b, 2011, 2013). Furthermore these satellite data have supplied critical information about the relationship of the ice density distribution with mesopause temperature and water vapour concentration (see, e.g. Hervig et al., 2009c; Russell et al., 2010; Hervig et al., 2015).

While PMCs emit thermal radiation, their infrared emissions are very difficult to observe due to the low ice particle volume density and the very cold polar summer mesopause temperatures. In fact, only three IR emission observations have been reported to date: that taken by CRISTA (Grossmann et al., 2006), by the SPIRIT (O'Neil et al., 2008) and those taken by MIPAS (López-Puertas et al., 2009). This technique has the advantages of being able to measure PMCs in dark conditions –thus providing a better spatial coverage–, and of being sensitive to the total ice volume density, regardless of particle size, and

hence including the very small particles.

In this previous paper (López-Puertas et al., 2009) we reported the detection of infrared emissions from PMCs taken by the Michelson Interferometer for Passive Atmospheric Sounding (MIPAS) instrument on board ENVISAT (Environmental Satellite), and provided further evidence of the water ice nature of the PMC particles. We also described the retrieval of the ice particle volume density and reported the analysis of the retrieved densities for 19-21 July 2005. In this paper we present the

global distributions (altitude, latitude and longitude) of the ice volume density measured by MIPAS for several days in each of the Northern (NH) and Southern Hemisphere (SH) seasons from 2005 until early 2012. We also analyse several aspects of the PMCs such as: (i) the mean altitude of the layer, the ice water content and their hemispheric dependence; (ii) the correlation of the ice volume density with the frost point temperature and with the water vapour concentration; and (iii) the diurnal variation of the ice volume density. MIPAS, as well as SOFIE, has the advantage of measuring the whole content of ice particles (all

sizes) in the mesosphere. Hence, a comparison with SOFIE observations is also shown.

## 2   MIPAS Measurements and Ice Density Retrieval

MIPAS is a high-resolution limb sounder on board the ENVISAT satellite, launched on March 1, 2002. It took measurements until 8 April 2012, when the Envisat satellite failed. MIPAS measurements covered a wide spectral range with a high spectral resolution, operating at 0.025 cm$^{-1}$ from 2002-2004 and 0.0625 cm$^{-1}$ from 2005 until the end of the mission. It also

operated with a high sensitivity, allowing measurement of most of the atmospheric emissions in the mid-infrared over a large altitude range (Fischer et al., 2008). MIPAS operated with a global latitude coverage (pole-to-pole) and performed measurements irrespective of day- or night-time. The instrument spent most of the time observing in the 6-68 km altitude range (the





**Table 1.** Days of MIPAS observations of PMCs in the different modes.

| Mode | Days |
|------|------|
| NLC | 20050719 20050720 20050721 20070704 20070705 20070714 20070715 20080705 20080706 |
|     | 20080707 20090105 20090106 20090107 20090705 20090706 20090707 20100104 20100105 |
|     | 20100106 20100703 20100704 20100705 20110109 20110110 20110111 20110708 20110709 |
|     | 20110710 20120104 20120105 20120106 |
| MA | 20050110 20050111 20050112 20050113 20051229 20051230 20060621 20060622 20061219 |
|     | 20061220 20061221 20070622 20070725 20070804 20071219 20071229 20080108 20080116 |
|     | 20080126 20080205 20080616 20080625 20080715 20080725 20080804 20081222 20090101 |
|     | 20090111 20090205 20090615 20090625 20090715 20090725 20090801 20090811 20091215 |
|     | 20091225 20100114 20100122 20100613 20100623 20100713 20100723 20100802 20100812 |
|     | 20110119 20110618 20110628 20110719 20110801 20110807 20111225 20120114 |
| UA | 20050121 20050122 20050722 20051231 20060623 20061222 20070620 20070621 20071220 |
|     | 20071230 20080109 20080117 20080127 20080206 20080622 20080716 20080726 20080805 |
|     | 20081223 20090102 20090112 20090119 20090120 20090206 20090616 20090626 20090716 |
|     | 20090726 20090802 20090812 20091220 20091230 20100109 20100117 20100618 20100628 |
|     | 20100718 20100728 20100807 20100817 20101225 20110104 20110114 20110124 20110130 |
|     | 20110131 20110201 20110623 20110703 20110714 20110727 20110804 20110812 20111220 |
|     | 20111230 20120109 20120124 |

nominal mode) but it also regularly observed at higher altitudes in its middle atmosphere (MA), noctilucent (NLC), and upper atmosphere (UA) modes (De Laurentis, 2005; Oelhaf, 2008).

In the MA mode, the spectra are available at limb tangent heights from about 20 km up to 102 km with a vertical sampling of 3 km. The UA mode ranges from about 42 km to 172 km, and has a vertical sampling of 3 km up to 102 km, and 5 km
5  above this altitude. The NLC mode is a variant of the middle atmosphere mode specifically tailored for measuring the PMCs during the summers (De Laurentis, 2005; Oelhaf, 2008). In this mode the spectra cover tangent heights from 39 km up to 78 km at 3-km steps; then from 78 km up to 87 km at 1.5 km steps, and from 87 km up to 102 km again in 3-km steps. MIPAS horizontal field of view (FOV) is approximately 30 km. The days of PMC measurements in the different observation modes are listed in Table 1, and a summary of the distribution of these days along the different seasons is shown in Table 2.
10  The method used for the inversion of PMC ice volume density from the MIPAS spectra has been described in López-Puertas et al. (2009). A brief excerpt is included here. The spectra analysed in this work were all taken with the optimized spectral resolution of 0.0625 cm$^{-1}$. The ice volume density was retrieved from the radiance profiles obtained by integrating the spectra from 730 to 950 cm$^{-1}$. The profiles were corrected for an offset variable in altitude, latitude and time.





**Table 2.** Distribution of MIPAS days of observation of PMCs per season[*].

| Year | NLC | | MA | | UA | | Total | |
|------|-----|-----|-----|-----|-----|-----|-----|-----|
|      | NH  | SH  | NH  | SH  | NH  | SH  | NH  | SH  |
| 2005 | 3   | -   | -   | 4   | 1   | 2   | 4   | 6   |
| 2006 | -   | -   | 2   | 2   | 1   | 1   | 3   | 3   |
| 2007 | 4   | -   | 3   | 3   | 2   | 1   | 9   | 4   |
| 2008 | 3   | -   | 5   | 6   | 4   | 6   | 12  | 12  |
| 2009 | 3   | 3   | 6   | 4   | 6   | 6   | 15  | 13  |
| 2010 | 3   | 3   | 6   | 4   | 6   | 4   | 15  | 11  |
| 2011 | 3   | 3   | 5   | 1   | 6   | 7   | 14  | 11  |
| 2012 | -   | 3   | -   | 2   | -   | 4   | -   | 9   |
| Total| 19  | 12  | 27  | 26  | 26  | 31  | 72  | 69  |

[*] For the NH season the days correspond to June-August of the listed year. For SH season the days correspond to December of the preceding year and January-February of the listed year.

The noise equivalent spectral radiance in this spectral region is about 20 nW/(cm² sr cm⁻¹), and the corresponding noise in the integrated radiances of a single scan is ∼60 nW/(cm² sr).

The ice volume density was retrieved from the spectrally-integrated radiance profiles using a linearly constrained least squares fitting, where the Jacobians were calculated using the KOPRA radiative transfer algorithm (Stiller et al., 2002). The inversion was constrained by a Tikhonov-type scheme (Tikhonov, 1963) using a squared first-order differences matrix to obtain a reasonably smoothed vertical profile of volume densities. The ice refractive indices were taken from Toon et al. (1994).

In this analysis we have included the following improvements and updates with respect to López-Puertas et al. (2009): (i) The more recent version 5 (5.02/5.06) of MIPAS L1b spectra has been used; (ii) an updated version of the temperature is used for the retrieval of ice density (see below); (iii) the altitude registration of the L1b spectra has been improved by using the information from the retrieved temperature and LOS (line of sight) instead of the engineering information included in the L1b files (von Clarmann et al., 2003; García-Comas et al., 2012); (iv) the offset correction of the integrated radiance profiles was improved by taking into account its altitude and latitudinal variations; (v) the ice density profiles were retrieved only for the scans with converged pressure-temperature profiles (no latitude/longitude interpolation was done); and (vi) due to a mistake in the calculation of the volume of the particles distribution, the volume densities presented here are nearly double those previously reported in López-Puertas et al. (2009).

The temperature and LOS required to retrieve the ice density have been inverted from the $CO_2$ emission near 15 $\mu$m, recorded in the same MIPAS band A as the PMC emission. Non-local thermodynamic equilibrium (non-LTE) emission was taken into account. The detailed description of the method and the characterization of the inverted temperature profiles are described in





García-Comas et al. (2012). The upgrades in the retrieval of the temperature used here (version vM21) and a validation of the results are reported by García-Comas et al. (2014). Briefly, these authors include an updated version of the calibrated L1b spectra in the $15\,\mu\mathrm{m}$ region (versions 5.02/5.06); the HITRAN 2008 database for $CO_2$ spectroscopic data; the use of a different climatology of atomic oxygen and carbon dioxide concentrations; the improvement of several aspects of the retrieval set-up

(temperature gradient along the line of sight, offset regularization, and the spectral apodization); and some minor corrections to the $CO_2$ non-LTE modelling as detailed by Funke et al. (2012). This version of MIPAS temperatures correct the main systematic errors of the previous version and have, in general, a remarkable agreement with the measurements taken by ACE-FTS, MLS, OSIRIS, SABER, SOFIE and the Rayleigh lidars at Mauna Loa and Table Mountain. In the region of interest here, however, there are still significant differences, with MIPAS polar summer mesopause temperatures differing by 5-10 K from

the other instruments, being warmer than SABER, MLS and OSIRIS and colder than ACE-FTS and SOFIE.

  Since MIPAS measures PMCs in emission, knowledge of the temperature of the ice particles is crucial. There is still disagreement about the temperature of the particles, particularly if they are warmer or colder than the ambient atmosphere. Using SOFIE measurements, Hervig and Gordley (2010) have found that the ice particles are about 5-20 K cooler than the ambient temperature. They suggested, however, that the V1.022 SOFIE $CO_2$ temperatures they used might have a warm bias of

5-10 K near the polar summer mesopause. Petelina and Zasetsky (2009), using infrared solar occultation measurements from the Atmospheric Chemistry Experiment (ACE) instrument, also found that the ice particles are cooler than the ambient temperature. They argue that this might be caused by inhomogeneities in the temperature along the instrument field of view, with the ice particles sensing only the cold(er) parcels, where they are present, while the gas temperature is representative of the whole (warmer) air mass along the FOV. Physical considerations, however, would suggest that the particles are warmer than

the surrounding gas because they will be heated up by absorption of radiation (Rapp and Thomas, 2006; Espy and Jutt, 2002). For example, for a particle distribution with a mean radius between 30 and 50 nm and an accommodation coefficient of 0.5, Rapp and Thomas (2006) found that the ice particles are warmer than the ambient gas in about 1 K at 80 km and 2 K at 90 km. Analogously, the model calculation of Espy and Jutt (2002), when applied to a normal distribution of ice particle size with a mean radius varying from 40 nm at 80 km to 15 nm at 90 km, gives a temperature increase of 0.7 K at 80 km and 2.7 K at

90 km. As suggested by these models, we applied a temperature correction of the emitting particles that varies linearly from 1 K at 80 km to 2 K at 90 km. In principle, MIPAS measurements should also be affected by the problem pointed out by Petelina and Zasetsky (2009). However, our observations do not support that finding. If we assume that the ice particles are cooler than the retrieved gas temperature we would obtain very high (and unreasonable) concentrations of ice particles (see Sec. 3).

The vertical resolution of the ice density vertical profiles, in terms of the half-width of the columns of the averaging kernel matrix depends on the observational mode. For the over-sampled NLC mode, it varies from ∼2.5 km at 81–82 km to ∼3 km at 86 km, and to 3.5–4 km at 90 km. For the middle and upper atmosphere modes (MA and UA), it is coarser, with values ranging from 3.5 to 4 km. The error in the absolute pointing is about 200 m.



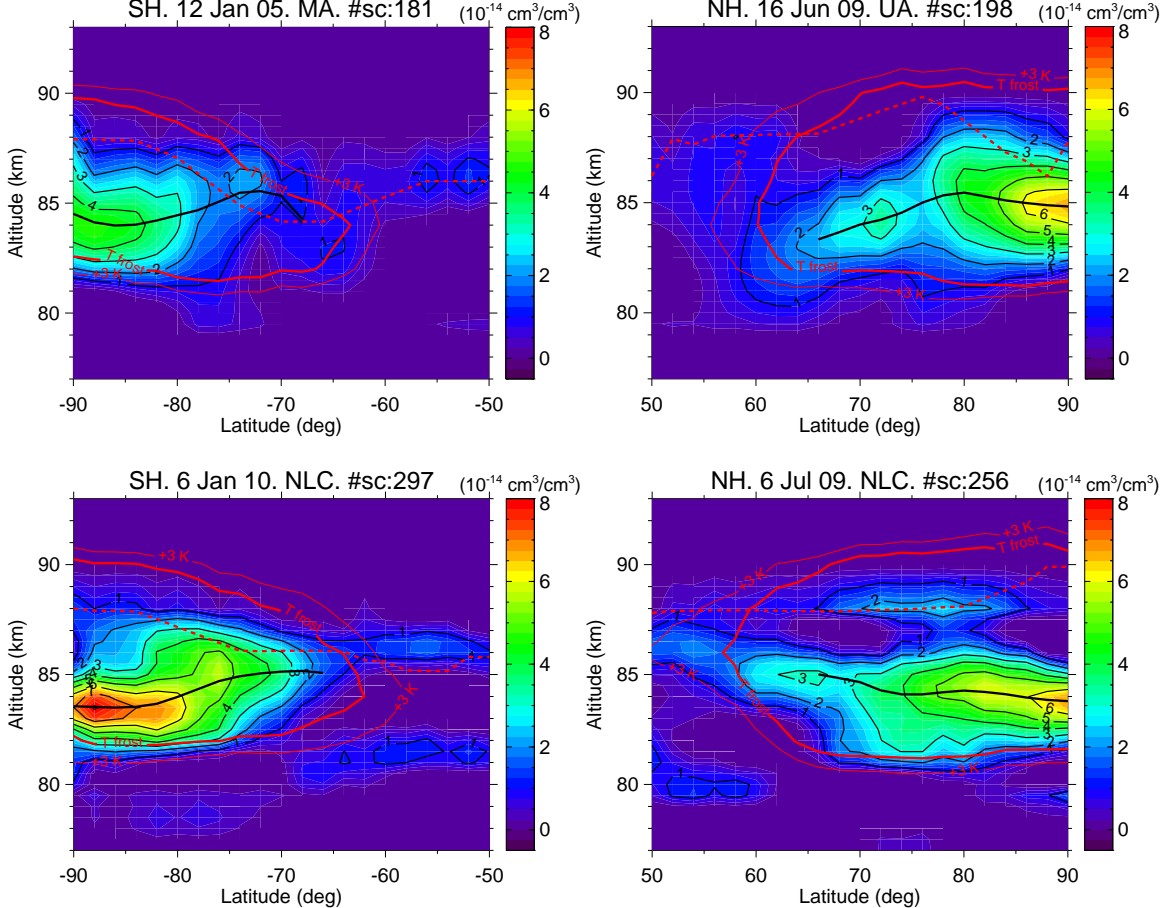

**Figure 1.** Zonal mean of ice volume density for four days, two in the Southern Hemisphere and two in the Northern Hemisphere as measured by MIPAS in different observation modes (MA, UA and NLC, see labels). The solid red lines indicate the frost point temperature (thick line) and frost point temperature plus 3 K (thinner line). The red dashed line is the mesopause as measured by MIPAS. The black solid line is an estimated mean altitude (weighted with the ice density to power of 4) of the PMC layer. The estimated noise error of the volume density plotted here is about $0.3 \times 10^{-14}$ cm$^3$/cm$^3$.

The averaging kernels shown in López-Puertas et al. (2009) are for the NLC mode measurements that have a sampling step (i.e. tangent altitude increment) of 1.5 km. For the MA and UA modes the averaging kernels are wider because of the coarser sampling of 3 km.

The random single profile error of the retrieved ice volume density is about 60%, including both the instrumental noise and the temperature noise error. The systematic error is about 25-30% and is mainly due to the temperature error in the summer mesopause region (García-Comas et al., 2014). More details of the retrieval of the ice volume density can be found in López-Puertas et al. (2009).



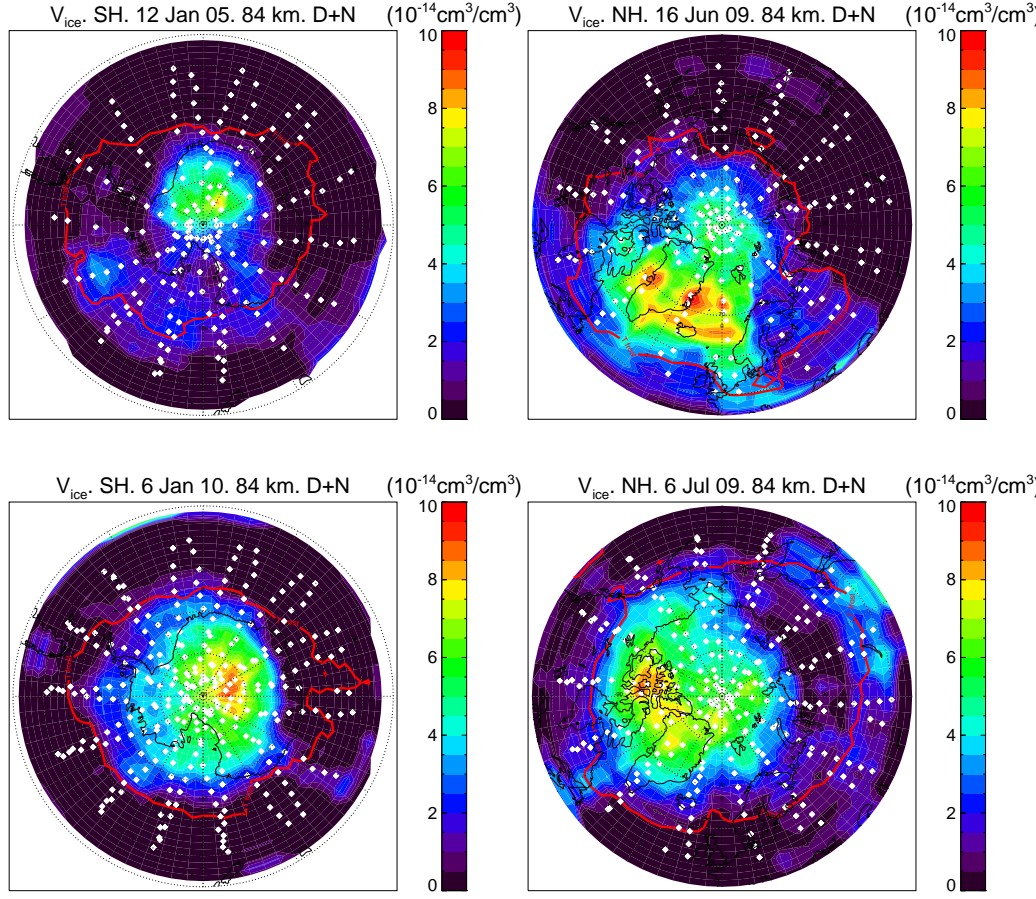

**Figure 2.** Latitude/longitude distribution maps of ice volume density at 84 km for the same days as in Fig. 1. The solid red lines indicate the frost point temperature. The diamonds represent the geolocations of the MIPAS measurements.

## 3   Ice volume density distributions

Figure 1 shows typical daily zonal means of ice volume density retrieved from MIPAS for four days in SH and NH summer seasons in different observation modes. The thick solid red line is the frost point temperature contour, and the red dash the mesopause altitude. The solid black line is an estimated altitude of the PMC layer (i.e., the altitude weighted with the 4th power of the density). Note that MIPAS measurements are sensitive to all ice particles, including those with small radius. Noise errors in these plots are about $0.3 \times 10^{-14}$ cm$^3$/cm$^3$. The PMCs are generally located at regions colder than the frost point temperature for almost all conditions. Note also the large variability in latitude and altitude of the ice density, particularly on 6 July 2009 (bottom right panel) where the PMCs reach latitudes as low as 60°N.

Anomalous low-altitude detection of weak PMCs (i.e., below ∼80 km and outside of the $T_{frost}$ region) could be due to the limb nature of the measurements. Emission from isolated clouds located in the LOS far away from the tangent point, and



hence at higher altitudes, can be measured and thus attributed to these lower tangent heights (see, e.g., Hervig et al. (2009b), their Fig. 11). Also, the FOV can affect the lower and upper boundaries of the layer. Hervig et al. (2009b) have shown that the bottom and top altitudes as measured by SOFIE, which has a FOV of 1.5 km, can be smeared out in about 1-1.5 km. These two effects, along with the temperature error, can explain why MIPAS observes occasional ice volume concentration at the bottom

5  of the layer at temperatures warmer than the frost point temperature, see bottom-left panel of Fig. 1 around 70°N.

The latitude/longitude distributions of ice volume density at 84 km for corresponding days are shown in Figure 2. As shown before for the zonal means, the PMC layer is almost always confined to regions with temperature below the frost point temperature. The variability of the latitude/longitude spread is also large. Although the PMCs are generally centred around the pole, they are sometimes far away (see top right panel in Fig. 2).

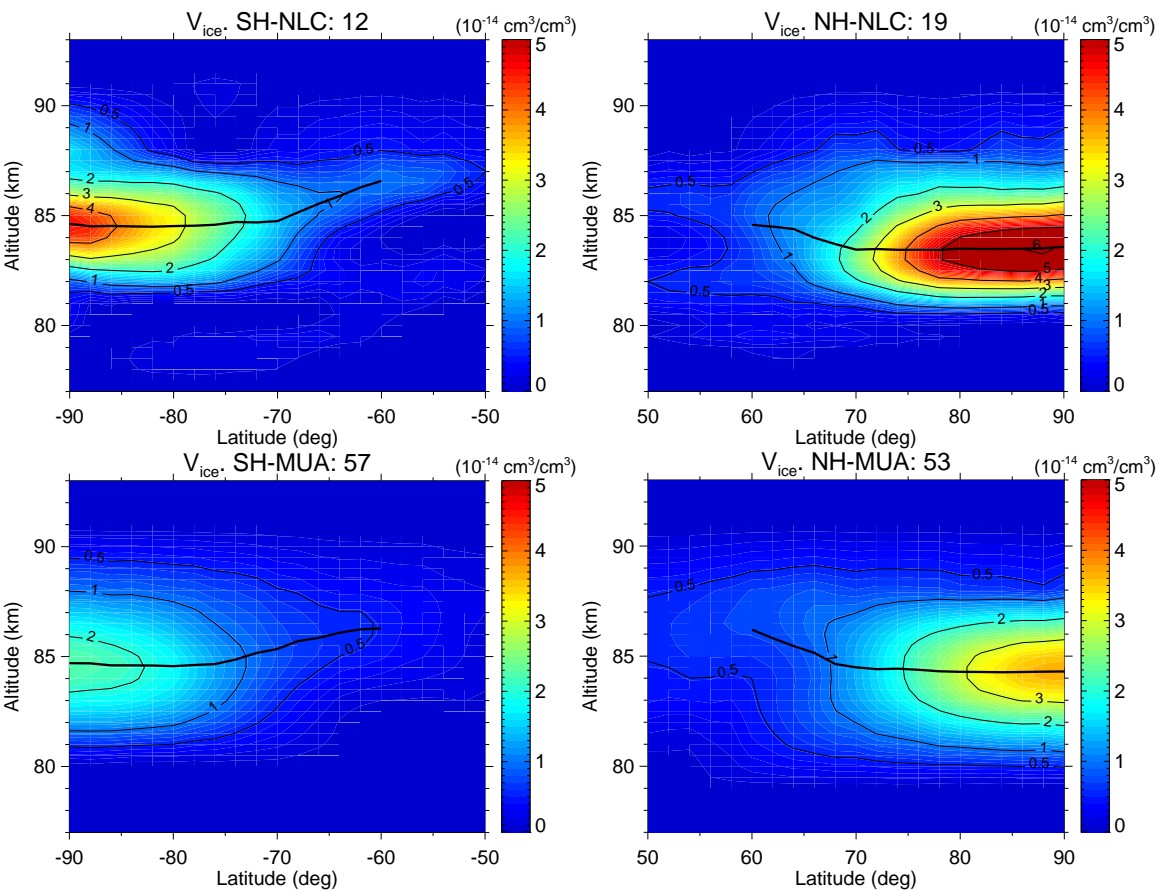

**Figure 3.** Zonal mean distributions of ice volume density for all measured days in the Southern (left panels) and Northern (right panels) hemispheres for the NLC (top panels) and for the MA+UA (MUA) (lower panels) MIPAS modes (see Table 2). The solid black line is an estimated mean altitude (weighted with the ice density to power of 4) of the PMC layer. The estimated noise error of the volume density plotted here is about $0.08 \times 10^{-14}$ cm³/cm³ and $0.04 \times 10^{-14}$ cm³/cm³ for the NLC and MUA measurements, respectively.



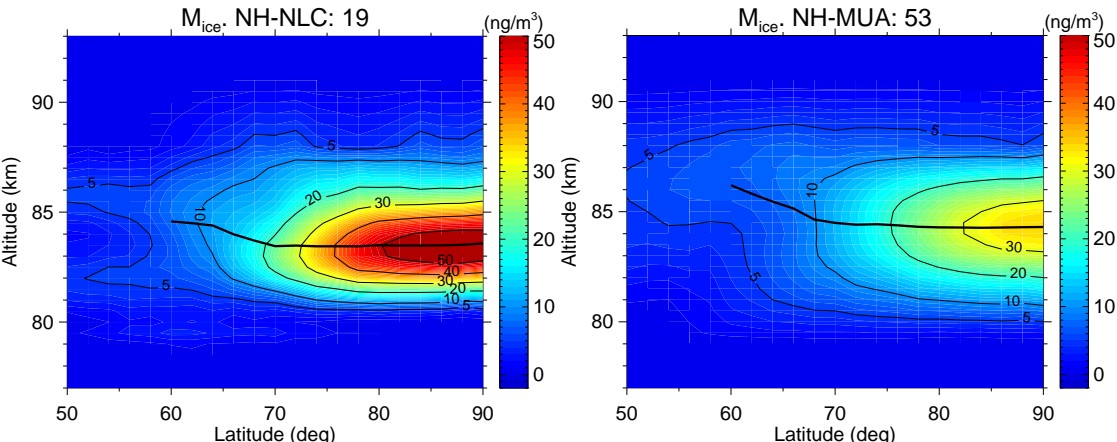

**Figure 4.** Zonal distribution of ice mass density for all measurements (see Table 2) in the Southern (left) and Northern (right) hemispheres. The solid black line is an estimated mean altitude (weighted with the ice density to the 4th power) of the PMC layer. The estimated noise error of the mass density plotted here is about 0.8 ng/m$^3$ and 0.4 ng/m$^3$ for the NLC and MUA measurements, respectively.

Figure 3 shows the zonal mean distributions of ice volume density averaged for all measured days in the Southern (left) and Northern (right) hemispheres for the NLC (top panels) and for the MA+UA (MUA) (lower panels) MIPAS modes (see Table 2). These distributions are analysed in detail later, but we describe the main features briefly here: 1) PMCs are confined to altitudes between around 81 km and 89 km with maximum concentrations around 84 km; 2) PMCs are confined to latitudes poleward

of about 60°, with increasing concentration towards the poles; 3) From these figures it is evident that the ice particles occur in higher concentrations in the NH, and that the ice layer is located at slightly lower altitudes in the Northern Hemisphere. These figures also show an apparent higher concentration for the measurements taken in the NLC mode than in the MUA mode. The NLC mode has a better vertical resolution, which leads to sharper temperature profiles (see García-Comas et al., 2014) and hence to sharper ice particle profiles and larger ice particle densities. However, not all the differences between the NLC and the

MUA modes can be attributed to the better vertical resolution of the former because they were taken in the summer on different days, with those of the NLC mode closer to the peak of the PMCs season.

## 3.1  Top altitude

Figure 3 shows that MIPAS observes significant abundances of ice up to about 88-89 km. A similar behaviour has been found in the SOFIE IR extinction measurements (Hervig et al., 2009b). This altitude is about 3-4 km higher than the average

maximum altitude of 84.4 km measured by the lidars. Hervig et al. (2009b) have shown for SOFIE that the vertical smoothing of the limb view geometry can cause an extension of the uppermost altitude of about 2/3 of the vertical resolution, i.e., 1.5-2 km for the MIPAS NLC observation mode. This, however, cannot fully explain that difference. The detection of PMCs by SOFIE and MIPAS at altitudes higher than the lidars is most likely due to the different sensitivities of the two techniques. While the lidar signal varies with $r^6$, the MIPAS (in IR emission) and SOFIE (in IR extinction) signals change with the total

ice volume density. As the ice particle size decreases towards higher altitudes (Baumgarten and Fiedler, 2008; Hervig et al.,



2009b; Pérot et al., 2010), MIPAS and SOFIE are then more sensitive than the lidars to higher altitude PMCs. The highest altitude of PMCs derived from MIPAS NLC mode measurements is about 89 km for the NH near 70° (Fig. 3b). This is slightly higher than that obtained by SOFIE of 86.8±2.1 km but agrees very well with the CARMA model prediction of 88.5±0.5 km (Hervig et al., 2009b, 2013). Thus, as pointed out by López-Puertas et al. (2009) and Hervig et al. (2009b), MIPAS and

SOFIE results are consistent with the current understanding of temperatures and water vapour distributions at these altitudes (Lübken, 1999), and the associated ice particles at high altitudes are likely to be related to polar mesosphere summer echoes (e.g., Rapp and Lübken, 2004). This has also been evidenced more recently by the concurrent observations from the ALOMAR wind (ALWIN) radar and measurements from SOFIE (Hervig et al., 2011).

### 3.2   Bottom altitude

The bottom altitude of the PMC layers measured by the lidar measurements in the Northern Hemisphere (at a latitude close to 70°) was found at 82.2 km. SOFIE obtained a slightly lower altitude of 81.6 km, which is within their mutual standard deviations (Hervig et al., 2009b). For the NH and similar latitudes MIPAS in its NLC mode (see Fig. 3b) gave an altitude of ∼81 km, slightly lower than SOFIE. Note, however, that we have not excluded any PMCs here, whereas in SOFIE those found below 79 km were excluded. The bottom altitude also changes rapidly with latitude from 65 to 75° (Fig. 3b); hence a few

degrees in latitude might also induce a significant change in the bottom altitude. Thus, in summary, we can conclude that they are in good agreement. It is also worth noting that the bottom altitude derived from the MUA modes, which have a coarser vertical sampling (3 km), is lower by about 1 km than that derived from the NLC mode (Fig. 3d). This is very likely due to the limb sounding geometry, as discussed above. The bottom altitude in the Southern Hemisphere is found to be located at about 1 km higher than in the NH (see Figs. 3a and 3c).

### 3.3   Concentration

As discussed above, MIPAS and SOFIE/AIM are the only two instruments whose ice concentration data are comparable because they both measure the total ice volume density, irrespective of the ice crystal size. Although it is not the aim of this paper to carry out a detailed comparison or validation, we include some comparisons here. First, we compare the maximum (peak) values of the PMC layer, and then we compare mean profiles for several seasons.

SOFIE measured ice mass densities at the altitude of maximum concentration, $z_{max}$, for the 2007 NH season that range from 0.1 to 80 ng m$^{-3}$, with a mean value of 14.2 ng m$^{-3}$ (Fig. 14a and Table 5 in Hervig et al., 2009b). These SOFIE measurements were taken at latitudes between 66°N, in the early season, to 74°N, towards the end of the season. MIPAS measurements for the 2005-2012 period at those latitudes have mean values of just above 20 ng m$^{-3}$ for the NLC mode and of ∼12 ng m$^{-3}$ (with a broader peak) for the MUA modes (see Figs. 4b and 4d, respectively), which agree well with SOFIE data for the 2007

NH season. As a result, the conclusion drawn by Hervig et al. (2009b) from SOFIE applies to the comparison of MIPAS with other measurements and models. That is, MIPAS ice mass densities are also significantly smaller than the lidar measurements taken at ALOMAR (69°N), that present an average value of 47.4 ng m$^{-3}$, and the lidar results reported by von Cossart et al. (1999), with ice mass ranging from 36 to 102 ng m$^{-3}$. MIPAS, as well as SOFIE, also measure thinner ice clouds than other





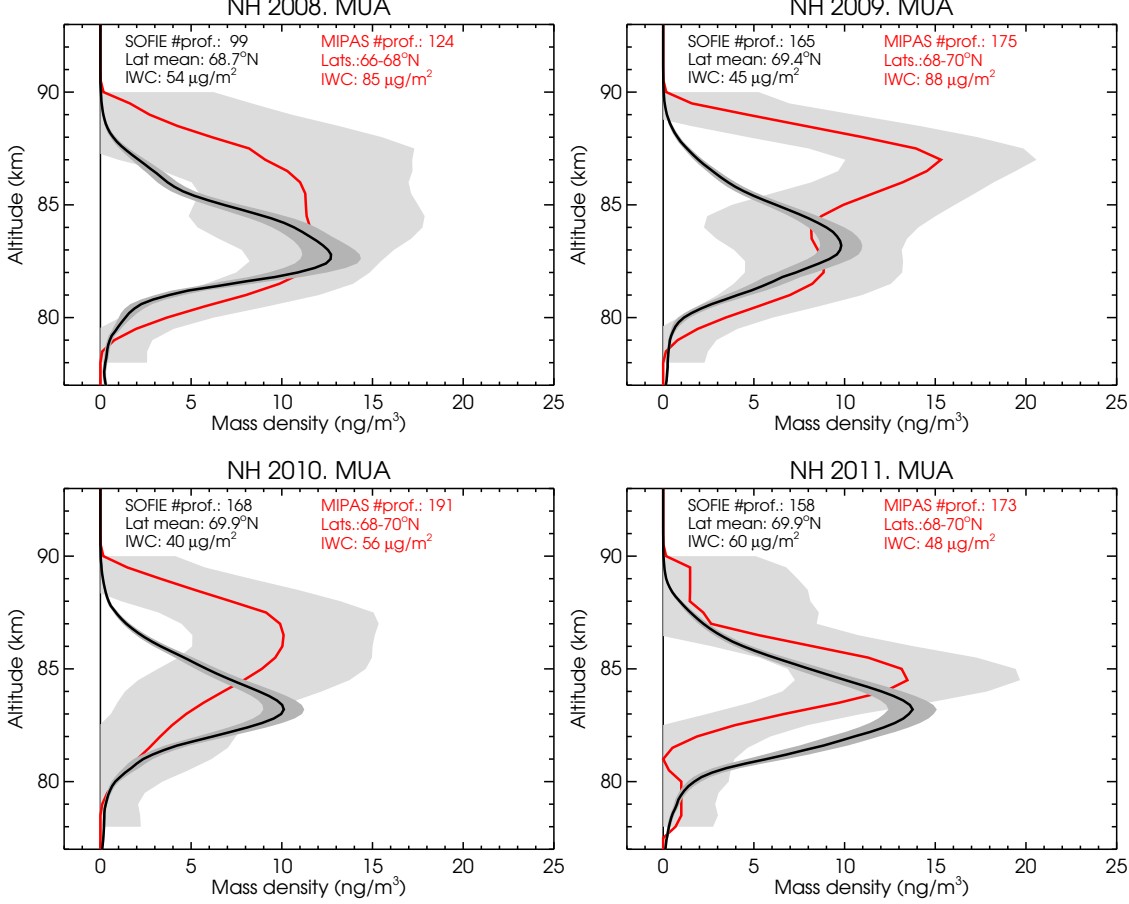

**Figure 5.** Comparison of the ice mass density of MIPAS MA and UA modes of measurements (see Table 1) with SOFIE v1.3 L2 data for the 2008 to 2011 period in the NH. The solid lines show the mean profiles, SOFIE in black and MIPAS in red. The shaded areas are the standard deviations divided by the square root of the number of profiles. The means of the integrated water column (IWC) are also shown.

IR instruments measuring the PMCs from space, e.g., HALOE (Hervig et al., 2003). Those differences can be explained, at least partially, by the larger sensitivity of MIPAS (and SOFIE) to the smaller particles (i.e., being sensitive to smaller amounts leads to lower mean concentrations). Another reason causing the differences could be, at least for the lidar observations, the averaging over the relatively larger atmospheric volumes sampled by MIPAS (and SOFIE).

5    Although a detailed comparison with the Community Aerosol and Radiation Model for Atmospheres (CARMA) (Rapp and Thomas, 2006) has not been performed, the results reported in Hervig et al. (2009b) suggest that MIPAS and CARMA are in agreement, at least for the 65-75° latitudes. A thorough comparison with the CARMA model, including higher latitude regions, is necessary but is beyond the scope of this paper.

Figure 5 shows a more detailed comparison between MIPAS and SOFIE ice mass densities, $M_{ice}$, for the coincident days and
10   latitudes in the NH season for the years with more coincident data: 2008-2011. The comparison is based on the mean profiles




for all days of measurements for each season/year for each instrument because of the large variability of the ice concentration (see, e.g. Fig. 1 in the case of MIPAS). The solid black lines represent the mean of the SOFIE measurements and the solid red line the MIPAS ice mean mass density. As discussed before, these figures also show that, in general, there is a very good agreement between the two instruments in the peak values of the layer (the 2009 NH season is an exception). However, above

about 85 km, MIPAS values are nearly double those measured by SOFIE (except in 2011). This MIPAS data feature, of large ice densities at high altitudes, can also be seen in the zonal mean distributions (Fig. 4, bottom right panel): the high values above 84 km extend from the North pole to fairly low latitudes, near 70°N or even lower. The same behaviour is seen in the MIPAS data for the SH (see left panels in Fig. 3). This seems to be a clear characteristic of MIPAS measurements but absent in SOFIE. We do not have a plausible explanation for this difference. In this region the ice particles are the smallest and it could

be that MIPAS is more sensitive than SOFIE to those particles. Another possible reason could be a negative bias of MIPAS temperature at those altitudes/latitudes which would result in a higher ice mass density, but such a bias present only in those localized regions seems unlikely. Note also that this zonal distribution of the ice density in MIPAS is consistent with the water vapour (gas phase) latitudinal distribution measured by MIPAS (see Fig. 10), since the depletion of water vapour near 60-70°N occurs at higher altitudes than near the North pole.

The integrated water column, which is written for both instruments in Fig. 5, is generally larger in the case of MIPAS, which essentially reflects the higher values in the ice mass densities of MIPAS at altitudes above ∼84 km discussed above. It is noteworthy, however, that MIPAS observations are in better agreement than SOFIE with model calculations carried out by Hervig et al. (2009c) (see their Fig. 5d).

### 3.4 $Q_{ice}$

We also show in Fig. 6 the zonal mean of ice volume density (similar to Fig. 3) but in units of ppmv, $Q_{ice}$; i.e., the partial concentration of water vapour if all the ice were to sublimate. For that conversion we used the pressure and temperature measured by MIPAS (García-Comas et al., 2014). As expected they show the same general behaviour as discussed above for the volume density. We note in general that the amount of water vapour in the form of ice ranges from 1 to 3 ppmv, although close to the North pole during the high season period (NLC) it can be as much as 6 ppmv. We think this is an important result

since, to our knowledge, the water ice content has not been measured at latitudes higher than ∼75°. Again these values are in good agreement with SOFIE measurements. Hervig et al. (2015) have shown time series of SOFIE $Q_{ice}$ for the 2007-2013 period for the Northern and Southern hemispheres. The NH mid-summer values range from 2 to 3.3 ppmv, which compare well with those shown in the right panels of Figure 6 at the latitudes of SOFIE measurements, ∼66°-74°N. Similarly, for the SH they show values spanning from 1.5 to 2.5 ppmv, also in good agreement with those of MIPAS shown in the left panel of

Figure 6. This point is discussed further in Sec. 6.





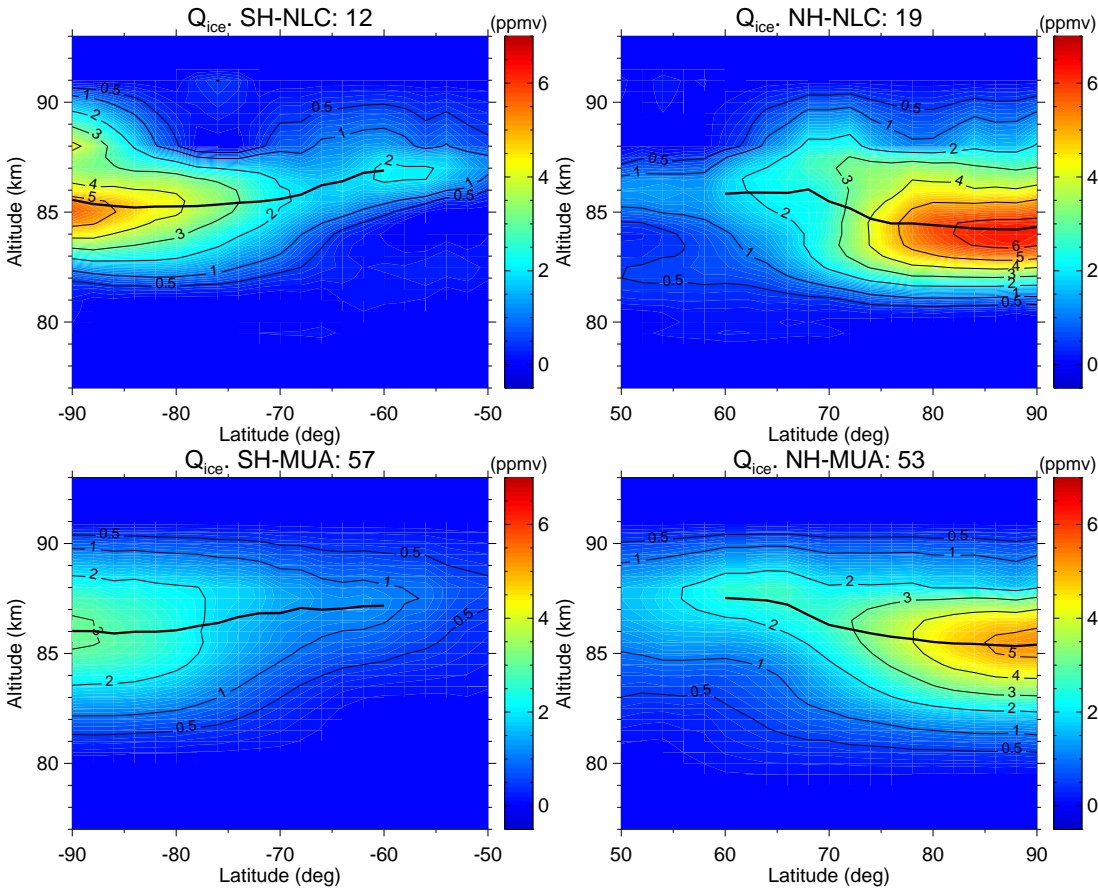

**Figure 6.** Same as Fig. 3 but in units of ppmv. The estimated noise error of the H$_2$O ice concentration plotted here is about 0.08 ppmv and 0.04 ppmv for the NLC and MUA measurements, respectively.

## 4  Altitude and column density of the PMCs

Figure 7 shows the mean altitude of the PMC layer for the SH (left) and the NH (right) seasons for all measurements. We observe that the mean altitude in the NH for the NLC mode is located around 83.5-84 km, while in the SH it is about 1 km higher (84.5-85 km). The fact that the mean altitude is higher (in ∼1 km) for the MA+UA modes is attributed to the coarser

5  sampling and to the broader vertical resolution in the retrieved temperature from these modes. The different temporal sampling of the NLC and MUA modes might also have an effect though. Hervig et al. (2013) have shown that PMCs are located higher at the beginning and the end of the season, and lower in the middle of the season. This coincides with our results since the NLC measurements are usually taken in the middle of the PMC season while MUA are taken earlier and later in the season. We should also note from Fig. 7 that PMCs tend to be located at lower altitudes near the poles, and at higher altitudes towards

10  mid-latitudes (both in NH and SH but more clearly in the latter).




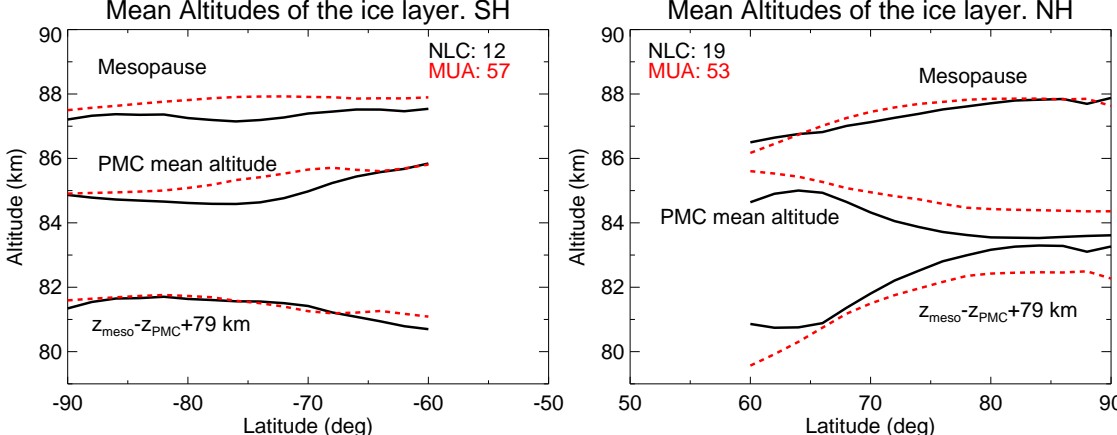

**Figure 7.** Mean altitudes of the mesopause ($z_{meso}$), of the PMC layer ($z_{PMC}$), and the difference $z_{meso} - z_{PMC}$ shifted 79 km, for the SH (left) and the NH (right) seasons for all measurements. The different colors indicate the results for the NLC and MA+UA (MUA) MIPAS observation modes (see Table 2).

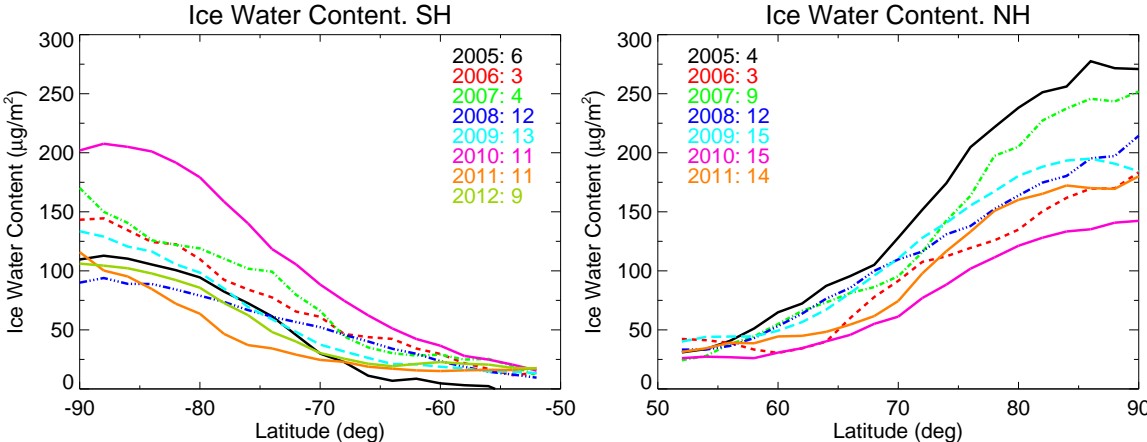

**Figure 8.** Latitudinal distribution of the ice water content of the PMC layers for the SH (left) and the NH (right) seasons for all measurements. The colors indicate the data for different years and the number of days measured per year (see Table 2).

Hervig et al. (2009b, 2013) reported an average value for the mean altitude of the PMC layer of 83.5 km for NH and 84.7 km for SH in SOFIE measurements, and of 83.3 km for the NH from the lidar measurements. The MIPAS mean value obtained here for the NH is very close to both measurements. Also, it is very much in line with SOFIE, locating the maximum of the layer about 1 km higher in the SH than in the NH.

5     Russell et al. (2010) carried out a multi-year analysis of the OSIRIS/Odin, SNOE, AIM, and SABER/TIMED data sets in the polar regions north (south) of 65°N (°S) and found that the mean PMC height is located 3.5 km±0.5 km below the mean mesopause height. In the case of SOFIE measurements, however, this difference is significantly smaller, in ∼1 km, for most





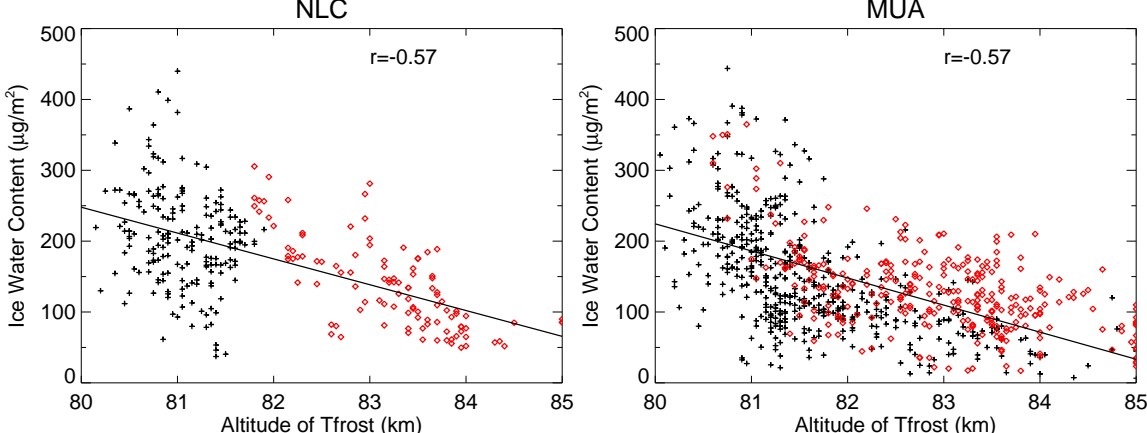

**Figure 9.** Correlation between the ice water content (IWC) and the altitude of the lower branch of the frost point temperature contour (see Fig. 1) for the data taken in the NLC (left panel) and MA+UA (right panel) observation modes in the NH (black pluses) and SH (red diamonds) PMCs seasons (see Table 2).

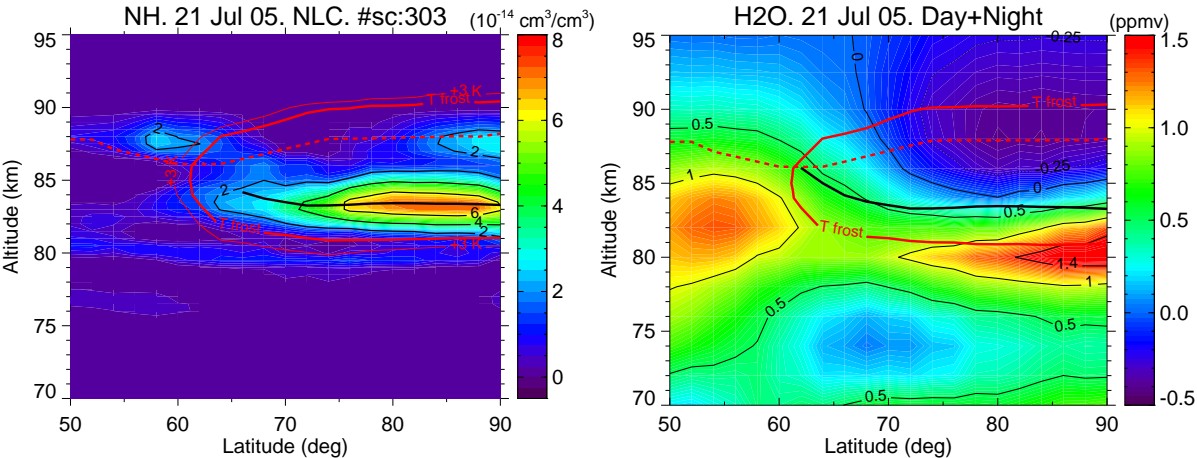

**Figure 10.** Zonal mean of the ice volume density (left) and of the $H_2O$ concentration anomaly (the mean profile has been subtracted) (right) for 21 July 2005. The solid red lines indicate the frost point temperature (thick line) and frost point temperature plus 3 K (thinner line). The red dashed line is the mesopause as measured by MIPAS. The solid black line is an estimated mean altitude of the PMC layer (see Sec. 4).

of the season, except around the middle of the season. We also looked at the difference between the mean PMC height and the mean mesopause height in the MIPAS PMC measurements (see Fig. 7). In general MIPAS observations are more in line with SOFIE observations than with the other instruments. For the case of NLC and MUA MIPAS observation modes in the NH near 70°N, the difference is about 2.5 km, smaller than the mean value of 3.5 km obtained for all instruments and closer

5   to the SOFIE value. It is worth noting that this altitude difference increases towards the North pole, more clearly in the case of the NLC mode (taken around the middle of the season) and reaching about 4 km. In the Southern Hemisphere the difference





between the mesopause and mean ice layer altitudes is even smaller than for NH, with values ranging between 2 and 2.8 km; again in better agreement with SOFIE than with the other instruments.

Figure 8 shows the latitudinal variation of the ice water content of the PMC layer for the SH (left) and the NH (right) seasons for all measurements. The figure shows clearly that PMCs are more abundant in the NH than in the SH, extending to lower latitudes in the NH. The main reason for this is the warmer polar upper mesosphere in the SH than in the NH, about a 10 K difference as measured by MIPAS (García-Comas et al., 2014). As shown in the zonal fields (Figs. 1 and 3), the ice column volume increases toward the poles. The large variability is also clearly visible, which in the case of MIPAS is attributable not only to the yearly changes but also to the daily variation because of the infrequent temporal sampling of MIPAS. The ice column is large for the NLC mode (not shown), in consonance with the zonal mean fields shown in Fig. 3. As mentioned before, this is probably due to the fact that the NLC measurements are taken around the middle of the season (see Table 1). The NH/SH ratio of the ice water content varies with latitude, ranging from about a factor of 2 near 60° to 1.4 near the poles, with a value of 1.7 near 70°, which is din very good agreement with the factor of 65% reported by Hervig et al. (2013) from SOFIE measurements.

## 5 Correlation of ice volume density with the frost point temperature

Figure 9 shows the correlation between the ice water content and the altitude of the lower branch of the frost point temperature contour (see Fig. 1) for the data taken in the different observation modes in the SH and NH PMCs seasons. The correlation is significant and shows that the PMC layers are denser and wider when the frost point temperature occurs at lower altitudes. Furthermore, the ice volume density is also anticorrelated with the mean altitude of the PMC layer (not shown), that is, that the denser PMC layers are located at lower altitudes and the thinner ones at higher altitudes, which is consistent with the behaviours shown in Figs. 7 and 8.

## 6 Correlation of ice volume density with $H_2O$ concentration

Hervig et al. (2015) suggest that, as opposed to other satellite instruments' like HALOE and MLS water vapour measurements, the vertical resolutions from SOFIE are well suited for the study of correlations between water ice and water vapour. This is also the case for MIPAS. Given the good latitude coverage of MIPAS (covering the whole polar region) and the fact that the instrument is able to measure the ice water content and the water vapour concentration simultaneously, we have looked at the zonal mean and latitudinal/longitudinal distribution of both quantities in the polar summer region. Fig. 10 shows a typical case (21 July 2005) of the zonal mean cross sections of the ice volume density (left) and the $H_2O$ concentration anomaly (right). Also Fig. 11 shows the latitude/longitude distributions for the $H_2O$ vmr at 90 km (top) and 80 km (bottom), and for the ice volume density at 83 km (middle). The water vapour concentrations have been derived from MIPAS high resolution spectra in the region around 6.3 μm. We used version v5r_h2o_M22 retrievals. The retrieval baseline is an extension to the lower mesosphere of the set-up described by Milz et al. (2005) with the updates described in von Clarmann et al. (2009). The main





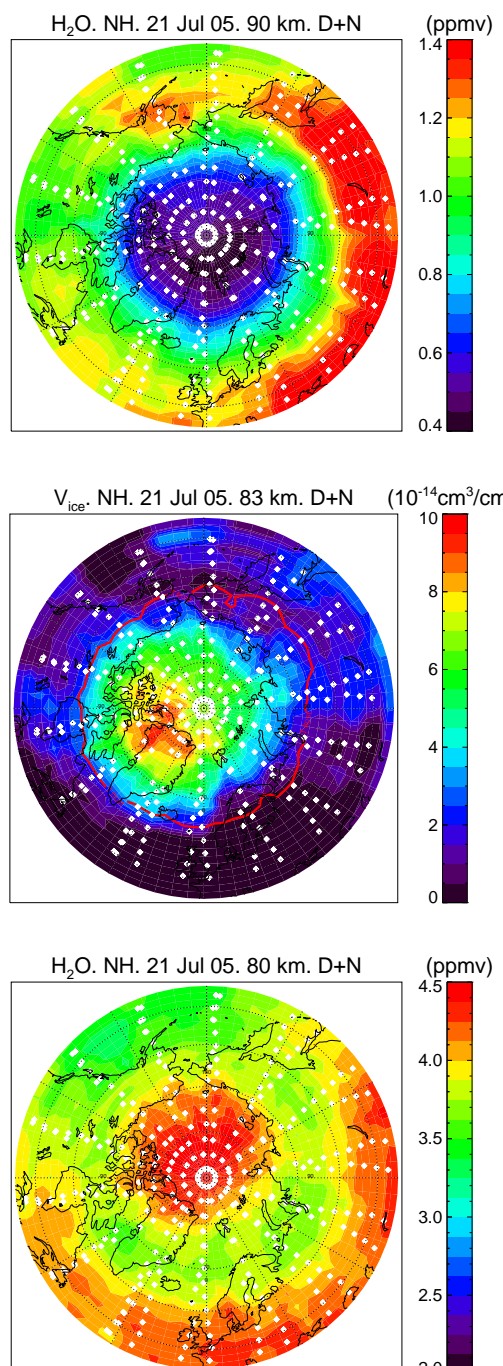

**Figure 11.** Latitude/longitude distribution maps of $H_2O$ vmr for altitudes of 90 km (top) and 80 km (bottom) (note the different scale) and of ice volume density at 83 km (middle panel) for 21 July 2005 (see Fig. 10).





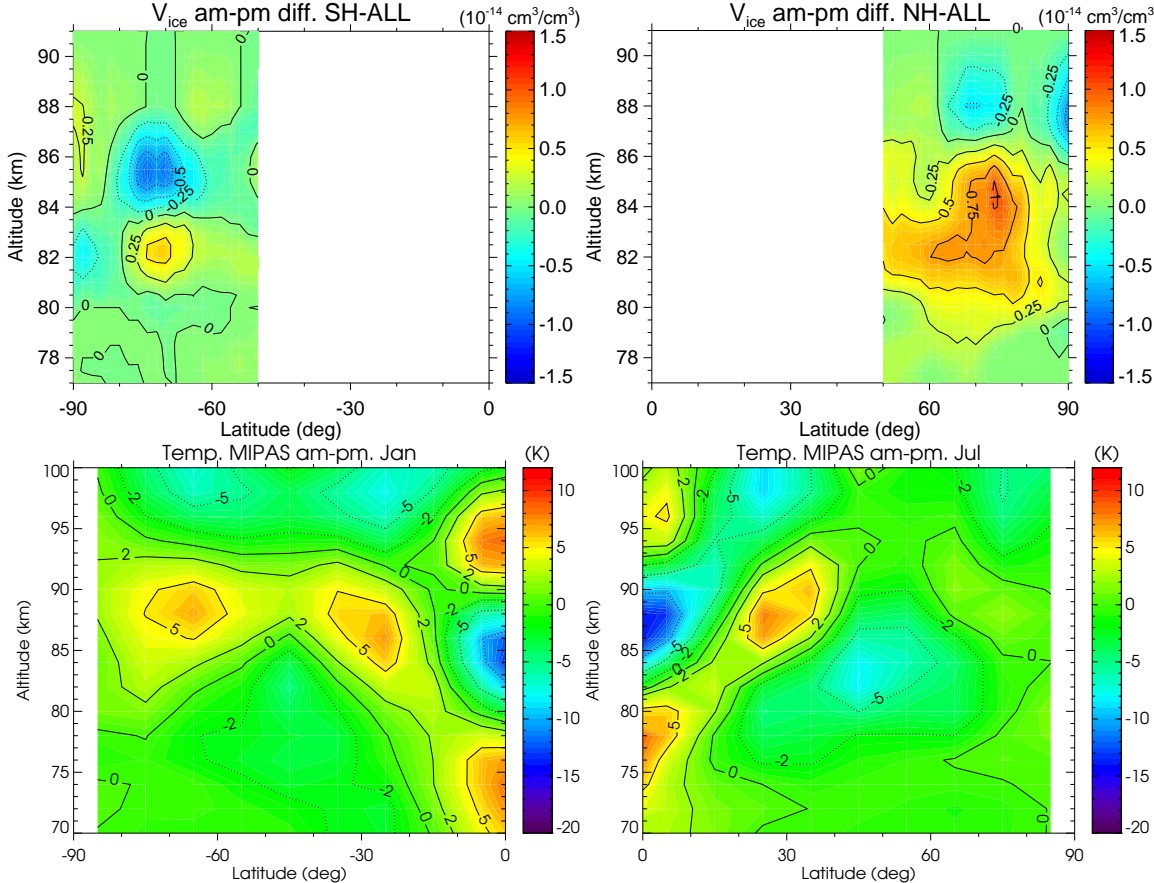

**Figure 12.** Top panels: zonal mean am-pm ice volume density differences for the SH (left) and NH (right) considering all measurements in each hemisphere. Bottom panels: zonal mean am-pm differences in temperature as measured by MIPAS for January (left) and July (right).

difference of this extension is the inclusion of non-LTE emission from the $H_2O$ vibrational levels, which are important above around 50 km (Stiller et al., 2012). Additional microwindows, covering stronger $H_2O$ $v_2$ spectral lines, are also included in order to increase the sensitivity in the upper mesosphere (García-Comas et al., 2012).

We can clearly distinguish three distinct altitude zones near the polar region. The region centred near the peak of the PMC

5 layer (∼83 km), where the ice volume density is largest; a few kilometers below, a hydrated region where $H_2O$ presents a relative maximum at latitudes northward of 70°N, more markedly seen in the bottom panel of Fig. 11; and a dehydrated region above the ice layer, around ∼ 90 km, where $H_2O$ exhibits a clear relative minimum (see top panel of Fig. 11). This global behaviour fits very well with the current picture we have about the PMCs, where sequestration of $H_2O$ in the gas phase to form ice leads to a drier atmosphere just above the ice layer, and where the sedimentation of ice and its subsequent sublimation

10 enhances the $H_2O$ gas phase abundance at ∼80 km. These features are more clearly observed in the latitude/longitude maps (Fig. 11), where the dry region at 90 km (top), the water ice layer at 83 km (middle) and the wetter $H_2O$ region at 80 km (bottom)





exhibit a very good latitude/longitude spatial correlation. This topic has been recently studied quantitatively by Hervig et al. (2015) by using SOFIE observation of ice content, water vapour and temperature at latitudes near 70°. They found that, in both hemispheres, the altitude of the peak of the dehydration regions is ∼1.8 km above the height of peak ice mass density, and the altitude of the peak of the hydration region is ∼0.3 km above the observed bottom altitude ice layer. Although no general

conclusion can be drawn from the single day of MIPAS data shown here, the location of the hydration region agrees well with SOFIE observations. The dehydration region, however, is found in MIPAS to be significantly higher than in SOFIE (see right panel in Fig. 10).

Hervig et al. (2015) also found that the column abundance of $H_2O$ in the gas phase is roughly equal in the dehydration and hydration layers, but less than that contained in the ice layer. MIPAS data also shows a similar feature, being more

pronounced at latitudes higher than those sounded by SOFIE. The right panel of Fig. 10 shows enhanced values of about 1.5 ppmv in the hydration layer and decreased by 0.5 ppmv in the dehydration region, while the $Q_{ice}$ peak is about 6 ppmv. A more comprehensive study, using all MIPAS data, should however be performed to confirm these findings. A further insight provided by MIPAS observations, with respect to SOFIE, is that this layer's structure is more pronounced at latitudes northernmost of 70°.

## 15  7  Diurnal variation of the ice volume density

The diurnal variation of the PMCs is an important factor to be taken into account when comparing the datasets for different PMCs. Several studies have shown that the IWC may have a very large diurnal variation at latitudes close to and equator-wards of 70°, mainly driven by tidal effects in the temperature and in the meridional advection at sub-polar latitudes (Stevens et al., 2010; Gerding et al., 2013). MIPAS measures PMCs at two local times, 10 am and 10 pm, and hence allows us to look at

the diurnal migrating variation (see García-Comas et al., 2016). Fig. 12 shows the diurnal differences (am-pm) of MIPAS ice volume density averaged over all measurements in the SH (left panel) and NH (right panel). The differences are larger in the NH, which are correlated with the larger concentrations in this hemisphere. The am-pm differences in the NH are larger at 60-80°, and reach a maximum value close to $10^{-14}$ cm$^3$/cm$^3$. This am enhancement is in line with the predictions of Stevens et al. (2010) but it is not as large as their calculations of a factor of 4.5 in the IWC at 69°N. At this latitude we find

a maximum daytime enhancement of about 60% in the volume density and 36% in the IWC. Note however that the IWC am-pm differences are also influenced by the slightly negative volume density difference at altitudes above 86 km. The vertically alternating increase at 84 km and decrease at 88 km in the am ice volume density indicates am clouds of lower altitude, also in agreement with Stevens et al. (2010).

The am-pm difference of ice volume density at 50-60° is 0.25-0.5·$10^{-14}$ cm$^3$/cm$^3$ at 81-87 km (Fig. 12). That corresponds

to an am/pm ratio lying between 1.5 at 86 km and 7 at 82 km. That indicates a significantly narrower and thinner pm cloud, in agreement with findings at sub-polar latitudes from Stevens et al. (2010) and Gerding et al. (2013), which mainly disappears below 84 km. The corresponding IWC am/pm ratio increases rapidly towards these lower latitudes and varies in the range of 1.5 to 2.8 at 50-60°.





These ice volume density differences are remarkably anti-correlated with the 10 am-10 pm differences in the kinetic temperature measured by MIPAS (bottom panels in Fig. 12) which are a good measure of the temperature perturbations due to the diurnal migrating tide. The negative am-pm difference at 80-85 km at latitudes below 80°N is well anti-correlated to the am-pm ice differences. Also, the temperature differences tend to be positive at higher altitudes northward of 65 °, which is well

reflected in the ice volume densities. Nevertheless, it is not possible to infer from this correlation alone, and without looking at wind fields, the extent to which these temperature perturbations affect the ice. Influence from other factors, like tidal effects on meridional advection (Gerding et al., 2013, see, e.g.) cannot be ruled out.

The anti-correlation between the diurnal variation of the ice density in the SH (upper left panel in Fig. 12) and that of temperature (lower left panel) is not ubiquitous. In this hemisphere, the negative temperature difference at 50-60°S and 80-84

km is weaker than in the NH. The diurnal positive ice change is correspondingly small but the ice volume density at these latitudes is also very low (less than $5 \times 10^{-15}$ cm$^3$/cm$^3$). The positive temperature difference above 85 km does anti-correlate with the negative am-pm ice concentration difference at 65-80°S. However, in contrast to the northern hemisphere, the am-pm temperature perturbation at 65-80°S is positive also at 80-85 km but so is the ice variation. This indicates that another factor affects the diurnal ice variation more significantly than temperature at those latitudes, at least below 84 km. Its effect results

in vertically alternating positive and negative changes that may lead to larger am-pm cloud altitude differences than in the northern hemisphere. A deeper analysis of this behaviour is beyond the scope of this paper and will be analysed in the future.

## 8    Conclusions

We have analysed the MIPAS IR measurements of PMCs for the summer seasons in the Northern and Southern Hemispheres from 2005 to 2012.

PMCs were measured in the middle IR in emission where, due to the small particle size, the signal is only affected by absorption and not by scattering. It is therefore sensitive to the total ice volume, including the very small ice particles, not generally sounded by the UV-VIS scattering instruments.

The measurements cover only a few days of the PMC season (varying from 3 to 15) but, on the contrary, have a global latitudinal coverage. In this way, MIPAS measurements show, for the first time, global latitudinal coverage (from 50° to the

pole) of the total ice volume density.

MIPAS measurements indicate mesospheric ice existing as a continuous layer extending from about ~81 km up to about 88-89 km on average and from the poles to about 55-60° in each hemisphere. These altitudes are in very good agreement with SOFIE measurements, with the lowest altitude being slightly lower (0.5 km) in MIPAS, and the uppermost altitude slightly higher (1-2 km) probably caused by the wider MIPAS field of view. This bottom altitude is also slightly lower than that derived

from lidars measurements but the uppermost altitude is significantly higher (4-5 km on average) than that obtained from lidar measurements. This indicates that both MIPAS and SOFIE instruments are sensing the small ice particles at the upper part of the PMC layer which are usually related to polar mesosphere summer echoes. This has also been proved recently by the concurrent observations from the ALOMAR wind (ALWIN) radar and measurements from SOFIE (Hervig et al., 2011).



The PMCs are very variable, both in space and time. On average, MIPAS measurements show that PMCs are confined to latitudes poleward of about 50–60°, with increasing concentration towards the poles. The water ice content in the PMCs measured by MIPAS at the latitudes of the measurements of SOFIE show, overall, a very good agreement, particularly at the peak of the layer. The water ice content observed by MIPAS is, in general, slightly larger, and also exhibits a larger variability,

probably caused by its smaller sensitivity. A distinctive feature, however, is that, in general, MIPAS shows significantly larger values in the region above ∼85 km, which can be twice those measured by SOFIE. In terms of ice water content, IWC, MIPAS are also generally larger than SOFIE values, principally caused by the larger concentrations above ∼ 85 km.

The ice concentration is larger in the Northern Hemisphere than in the Southern Hemisphere. The ratio between the IWC in both hemispheres is also latitude-dependent, varying from a NH/SH ratio of 1.4 close to the poles to a factor of 2.1 around 60°.

This also implies that PMCs extend to lower latitudes in the NH.

We have found that the mean altitude of the PMC layer in the NH for the NLC mode of MIPAS observations is located around 83.5-84 km, while in the SH it is about 1 km higher (84.5-85 km). This hemispheric asymmetry is in very good agreement with SOFIE observations (Hervig et al., 2013). For those MIPAS observations taken in the middle and upper atmosphere modes (MA and UA), the mean altitude is higher (in ∼1 km). This difference is attributed to the coarser sampling and to the broader vertical

resolution (particularly in the retrieved temperature) and also to their different temporal sampling since the NLC measurements are usually taken in the middle of the PMC season while MUA are taken earlier and later in the season. A very clear feature in MIPAS observations is that PMCs tend to be at higher altitudes as we move away from the polar region (in both hemispheres), particularly equator-wards of 70°.

MIPAS observations show that the difference between the mean PMC height and the mean mesopause height is about 2.5 km

in the NH near 70°N. This is smaller than the mean value of 3.5 km obtained for several instruments by Russell et al. (2010) and closer to the SOFIE value (Hervig et al., 2013). MIPAS also shows that this altitude difference increases towards the North pole, reaching a value close to 4 km. In the Southern Hemisphere this difference is smaller than for the NH, with values ranging between 2 and 2.8 km; again in better agreement with SOFIE than with the other instruments.

The anti-correlation between the ice water content and the altitude of the lower branch of the frost point temperature contour

is significant in MIPAS observations and shows that the PMC layers are denser and wider when the frost point temperature occurs at lower altitudes.

The simultaneous observations of PMCs and water vapour of MIPAS have shown that the PMC layers are surrounded by a hydrated region below and a dehydrated region above. These regions are more pronounced towards the poles, particularly at latitudes northernmost of 70°N. This global behaviour fits very well with the current picture we have about the PMCs

where sequestration of $H_2O$ in the gas phase to form ice leads to a drier atmosphere just above the ice layer, and where the sedimentation of ice and its subsequent sublimation enhances the $H_2O$ gas phase abundance at ∼80 km. The analysis of a single day of water vapour and PMCs measurements have shown that the location of the hydration region agrees well with SOFIE observations (Hervig et al., 2015). The dehydration region, however, is found to be significantly higher in MIPAS than in SOFIE. Further, as for SOFIE, measured near 70°, MIPAS shows that the column abundance of $H_2O$ in the gas phase is



roughly equal in the dehydration and hydration layers, but less than that contained in the ice layer. MIPAS observations at latitudes north of 70° show that this layering structure is more pronounced.

Finally, MIPAS observations, which are taken at 10 am and 10 pm, also show a diurnal variation in the ice volume density, with larger concentration, and slightly lower altitudes, at am than at pm, in line with the model predictions of Stevens et al.

5 (2010). This diurnal variation is anti-correlated with corresponding differences in temperature in the northern hemisphere, suggesting that it is driven by the temperature migrating diurnal tide, but effects from other factors cannot be ruled out. In the Southern Hemisphere, the lack of anti-correlation with temperature suggests the impact of an additional factor below 84 km.

*Acknowledgements.* We thank Patrick Espy for useful discussions about the temperature of the ice particles. We thank the SOFIE team for providing data, taken from http://sofie.gats-inc.com/sofie/index.php. The IAA team was supported by the Spanish MICINN under project

10 ESP2014-54362-P and EC FEDER funds. MGC was financially supported by the MINECO under its 'Ramón y Cajal' subprogram.



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
