# Peer review of "Measurements of Global Distributions of Polar Mesospheric Clouds during 2005-2012 by MIPAS/Envisat"

_Atmospheric Chemistry and Physics, 2016_

## Referee Comment (RC1) · Anonymous Referee #2 · 1 Apr 2016

Polar mesospheric cloud ice volume density is retrieved from MIPAS spectra. The global distribution of total ice volume density is shown for the first time for several days in both the Northern Hemisphere and Southern Hemisphere. This work is generally in agreement with previous work. The analysis is thorough and appropriate for publication in ACP following some revisions.

Major Comments:

Given the limited amount of data used from MIPAS, it would be useful to better quantify the observed variability in PMC properties throughout the paper. For example, what is the range of "top altitudes" from MIPAS? If there is a high degree of variability, the mean calculated from 12 or 19 days is likely insufficient to converge on the "true" mean.

I would check Figure 5, as I would expect better agreement with SOFIE based on the rest of your analysis. In 2009, for example, the average of 175 MIPAS profiles at 87 km is ~15ng/mˆ2, compared to ~2ng/mˆ2 from 165 profiles from SOFIE. If this figure is correct, then I think the large differences and variability compared with SOFIE call into question the entire analysis. It also looks like you may even be seeing ice above 90 km in Figure 5 and are simply setting these values to 0. Also, Figure 4 and Figure 5 do not seem to be in agreement. 2009 and 2010 show a peak in mass density above 87 km that is not represented at all in Figure 4.

Comments:

General: make sure you define each acronym once, the first time it appears.

Line 25: can't temperatures be lower that 150K?

The paragraph beginning at line 35 provides little information except to say that PMCs have been studied. What did these papers show?

Line 55: This paragraph could be reduced to say that similar results were found by Stevens and Hervig [2014] using SBUV.

Line 70: What do you mean by "the responses"? Are you saying that the 27-day solar cycle somehow accounts for long term PMC trends?

Titles of Figure 4 are not consistent with figure caption. It looks like the figure caption is wrong.

In Figure 5, put the red line on top of the shading.

What is the reasoning for showing a single day in Figures 10 and 11? Wouldn't it make more sense to do this analysis using all the data, so you could more easily compare with previous work?

Why do you show latitudes equatorward of 50° in Figure 12? Also, the temperature anomalies do not seem to correspond to the anomalies in ice volume density. Maybe

this is because you are comparing January and July differences in temperature to full season differences in ice volume density. Also, how do your results in Figures 12a and 12b affect your comparison to SOFIE in Figure 5? Would it make sense to only compare am or pm to SOFIE? SOFIE observes sunrise in the NH summer and sunset in the SH summer.

Figure 9 seems to show two distinct populations for the NH and SH. I don't think it makes sense to do a regression analysis of both hemispheres. Looking at the left panel, it seems that there is a strong linear trend in the NH, but in the SH, ice water content seems independent of frost point altitude. Maybe expand your analysis to discuss hemispheric differences and compute the correlation in each hemisphere separately.

Line 217: There is no 70°N in the bottom-left panel of Figure 1

Figure 2: Any thoughts on what drives the zonal variability observed here (i.e., planetary waves such as the 2-day or 5-day wave)? See Siskind, Nielsen, and Merkel

Line 261: What is the standard deviation for MIPAS? You talk about Figure 4 like it is a 4 panel plot, but it only has 2 panels

Line 295: Bardeen et al. [2010] has done this exact analysis.

Line 355: What lidar measurements, and at what latitude?

Line 390: I don't understand how this result is consistent with Figures 7 and 8, which have nothing to do with the bottom altitude.

Line 460: This anti-correlation doesn't seem apparent to me.

Line 513: Didn't you show this in Figure 1 and 3?

Minor Changes: Line 75: change "very little sensitive" to "not very sensitive" Line 84: "icy" to "ice" Line 85: remove comma after "observations" Line 86: remove "the" Line 87: This suggests that the advantages can measure. Change to "advantages that make

it possible to measure..." Line 89: change "including" to "include" Line 90: change "in emission" to "through emission" Line 108: change "which allows to measure" to "allowing it to measure" Line 122: change "along" to "within" Line 123 & 147: Maybe you mean "retrieval" instead of "inversion"? Line 127: change "off-set" to "offset" Line 158: "correct" should be singular since it refers to "version" Line 178: change "in" to "by" Line 184: change "pointed by" to "pointed out by" Line 185: "particle" should be plural Line 202: change "red dash the" to "red dashed line is the" Line 206: change to "for almost all conditions" Line 228: change "From these figures is" to "From these figures it is" Line 229: "occur" should be singular as it refers to "concentration" Line 249: change "altitudes" to "altitudes" Line 265: change to "might also induce a" Line 306: should read "there is very good" Reword sentence beginning on Line 377. Reword sentence beginning on Line 394. Line 447: remove "am" Line 433: change "decreased" in to "a decrease of" Line 493: change "being the lowest altitude" to "with the lowest altitude being" Line 494: add a comma after "(1-2 km)" Line 496: "lidars" should be "lidar" Line 508: change sentence to "Ice water content from MIPAS is also generally..." Line 518: change "in" to "by"

Bardeen, C. G., O. B. Toon, E. J. Jensen, M. E. Hervig, C. E. Randall, S. Benze, D. R. Marsh, and A. Merkel (2010), Numerical simulations of the three-dimensional distribution of polar mesospheric clouds and comparisons with Cloud Imaging and Particle Size (CIPS) experiment and the Solar Occultation For Ice Experiment (SOFIE) observations, J. Geophys. Res., 115, D10204, doi:10.1029/2009JD012451. Merkel, A. W., R. R. Garcia, S. M. Bailey, and J. M. Russell III (2008), Observational studies of planetary waves in PMCs and mesospheric temperature measured by SNOE and SABER, J. Geophys. Res., 113, D14202, doi:10.1029/2007JD009396. Nielsen, K., D. E. Siskind, S. D. Eckermann, K. W. Hoppel, L. Coy, J. P. McCormack, S. Benze, C. E. Randall, and M. E. Hervig (2010), Seasonal variation of the quasi 5 day planetary wave: Causes and consequences for polar mesospheric cloud variability in 2007, J. Geophys. Res., 115, D18111, doi:10.1029/2009JD012676. Siskind, D. E., and J. P. McCormack (2014), Summer mesospheric warmings and the quasi 2 day wave, Geophys. Res. Lett., 41,

717–722, doi:10.1002/2013GL058875.

---

## Referee Comment (RC2) · Anonymous Referee #1 · 1 Apr 2016

Overall recommendation: minor revisions

General comments: overall quality of the discussion paper This study introduces and describes a PMC dataset from MIPAS IR emission observations, shortly presents the retrieval method that was already published in a previous paper, provides evaluation of its quality by comparison of MIPAS ice mass density and cloud altitude to AIM SOFIE observations, and discusses MIPAS cloud properties in relation to previous findings. The important advantages of an IR emission dataset like MIPAS compared to other (UV and/or VIS) remote sensing datasets are: o that observations are available during day and night o but also that retrieved cloud properties are independent of the highly uncertain PMC particle size distribution, which has to be assumed for the retrieval of

cloud properties of many other satellite datasets. As such, this paper represent a substantial contribution in the form of new data to scientific progress within the scope of Atmospheric Chemistry and Physics. While the paper would profit from proof-reading by a native English speaker, there are also weaknesses in the discussion of the MIPAS measurement threshold, the discussion of the disagreement with SOFIE ice mass density profiles above 84km (Figure 5), and the discussion of MIPAS diurnal variations. A more thorough evaluation of the dataset could be achieved by additional comparison to another PMC dataset that also offer polar coverage, however, that would probably lengthen this paper too much. On the other hand, the paper can be shortened by emitting results that are not discussed in detail, for example Section 5. The conclusion should be more quantitative when summarizing the agreement with SOFIE. In summary, I think that all the here mentioned weaknesses can be resolved using the existing dataset, so I recommend this paper for publication in ACP with minor revisions.

Specific comments: individual scientific questions/issues 1. You compare the MIPAS PMC dataset to SOFIE, which observes PMCs at just one latitude each day. This latitude is slowly varying during the PMC season, but basically this restricts your comparison to a narrow latitude range. Have you considered comparing your dataset to other satellite observations, e.g., from CIPS, OSIRIS, SCIAMACHY, . . .? 2. You retrieve the ice volume density as it is independent of the assumption of the particle size distribution, which is considered uncertain. Have you considered retrieving the particle size and number density, using the same assumption that the SOFIE team uses? 3. Do you have plans for making this dataset publicly available? 4. Your introduction should state clearly which SOFIE version you are using. I have found this information in the figure caption of Figure 5, but it belongs in the introduction. 5. P3 L24-25, also P8 L5: I'm missing a more careful discussion of the MIPAS measurement threshold as done by Hervig et al., Interpretation of SOFIE PMC measurements . . . (2009). Some points regarding the SOFIE detection threshold and how it affects the radius retrieval from that paper: - The SOFIE ice detection threshold corresponds to Mice~0.06 ngm3. - It is important to note that particle size is only determined when the extinction $\beta(1.037)$

is above the noise. - Although SOFIE can rarely determine particle size for the most tenuous clouds, size is characterized over the dominant range of measurements. 6. P3 L30: It also operated with a high sensitivity – How do you define high sensitivity? 7. P6 L8-10: Please comment on how a potential 5-10K cold or warm bias affects your retrieved clouds. Do you believe The MIPAS temperature measurements more/less than those of other instruments? 8. Regarding all figures plotted vs. altitude: shouldn't the vertical axis be tangent altitude, not altitude? 9. Do you have enough lines of sight through one cloud volume that you could apply a tomographic algorithm? Such an algorithm has the advantage that it solves the problem of clouds in the fore- or background being assigned anomalously low tangent altitudes. You mention this problem on P8/9. Due to this problem SOFIE discards all clouds below a tangent altitude of 79 km. Examples for tomographic algorithms applied to PMC data: Hultgren et al., First simultaneous retrievals of horizontal and vertical structures of Polar Mesospheric Clouds from Odin/OSIRIS tomography, 2013; Hultgren and Gumbel, Tomographic and spectral views on the lifecycle of polar mesospheric clouds from Odin/OSIRIS, 2014. 10. P9 L8-9: Please comment on why in the upper left panel of Figure 2, the clouds fill only a small portion of the region where the temperature is below the frost point temperature, while on the lower left panel (also SH) the cloud coverage seems to agree much better with the frost point temperature boundary. 11. P9 L8-9: Please comment about the possibility that you see the effect of 5-day planetary wave activity in Figure 2. You could also comment about the possibility to use your dataset to help track the effect of space shuttle exhaust on PMCs, e.g., Stevens et al., Antarctic mesospheric clouds formed from space shuttle exhaust, 2005; Stevens et al., Bright polar mesospheric clouds formed by main engine exhaust from the space shuttle's final launch, 2012; Stevens et al., Polar mesospheric clouds formed from space shuttle exhaust, 2003. 12. P11 L12-13: reformulate sentence, e.g.: Using the NLC mode data at similar NH latitudes, we derive a mean bottom altitude of $\sim$81 km, slightly lower than that of SOFIE. But I think "similar NH latitudes" is too imprecise. It is not clear how exactly you choose the MIPAS latitude for comparison to SOFIE: do you use one mean latitude value, or a daily changing latitude value based on the changing SOFIE latitudes? Please describe your method better. 13. P11 L13-14: Note, however, that we have not excluded any PMCs here, whereas in SOFIE those found below 79 km were excluded. – You're not comparing apples and apples: what happens when you treat MIPAS observations like SOFIE did? Do you then get a better agreement (as expected)? 14. P11 L28: at those latitudes – please be more precise: what is your coincidence criterion? It may be worth showing a histogram of SOFIE and MIPAS ice mass. This would be helpful in convincing the reader of the nice agreement. 15. P12 L1-3: I don't understand your explanation why MIPAS and SOFIE are expected to observe less ice mass density than the lidar: if MIPAS and SOFIE are able to observe a BIGGER population of the total ice mass by ALSO observing the smaller particles (that the lidar does not observe), shouldn't the resulting ice mass density be BIGGER than the lidar ice mass density? 16. P13 L9-10: ice particles are the smallest and it could be that MIPAS is more sensitive than SOFIE to those particles. – Here you argue that a more sensitive instrument should result in higher values of ice mass density. On the previous page you have argued the opposite: that the larger sensitivity of MIPAS to the smaller ice particles than lidar will lead to lower mean ice mass density values. It makes the impression as if you are contradicting yourself. 17. If you haven't done that yet, I would suggest talking to Mark Hervig directly about possible reasons for the disagreement in ice mass density above 84km as seen in Figure 5. Please also discuss possible reasons for this disagreement in more detail, e.g., the role of geophysical differences. 18. P17 L15-17: The discussion of Figure 9 is very short (2 sentences) and contains only a description of Figure 9. Do you have an overall point you want to get across with Figure 9? Is this a new result or do you show this to relate MIPAS observations to previous studies (which?)? Otherwise, please consider omitting Figure 9 and Section 5. 19. Last paragraph of Section 6: the discussion about column abundance of gas phase H2O around 70° is not supported well by Figure 10, which shows the gas-phase H2O vs. altitude and latitude. 20. P20 L1: exhibit a very good latitude/longitude spatial correlation – I don't agree: while the dehydrated "hole" in H2O at 90 is neatly centered

on the pole, the clouds' center of mass is shifted towards northern Greenland, and at 80km the center of mass of the hydrated region is over the northern Pacific. I wouldn't call this "very good latitude/longitude spatial correlation", but expect a comment on this "rotation". 21. P20 L4-5: I don't agree with your statement that the location of the hydration region agrees well with SOFIE observations. From Figure 10 it looks like the MIPAS peak altitude of the hydration region is at 80km, whereas the bottom of the PMC layer is at 81 km. If anything, then the MIPAS peak in hydration lies BELOW the bottom of the PMC layer. Or do you also count the dark blue shading as PMC? Then I would agree. But for that it would be useful to know if these dark blue PMC observations are above the noise threshold. 22. P20 L9-10: What do you mean with being "more pro-nounced"? That the ice layer contains even less $H_2O$ than the hydration/dehydration layers? At higher latitudes the dehydration (-0.3 or -0.4 ppmv maybe) looks much less pronounced than the hydration (1.4 ppmv), which does not agree with the SOFIE re-sults that they are roughly equal. But again: my misunderstandings could be solved by showing a plot of the $H_2O$ column abundances. 23. Section 7 (Diurnal variation of ice volume density, Figure 12) is lacking clarity and not convincing: o P20 L26-28: as you write, there is an altitude difference between the morning/evening clouds, and you note that this altitude difference leads to the altitude bipole structure in Figure 12. Is it possible to correct for the altitude difference in order to get rid of the bipole structure? o P21 L1-2: These ice volume density differences are remarkably anti-correlated with the 10 am-10 pm differences in the kinetic temperature measured by MIPAS – I don't agree: there is a positive difference in T at 60-70S and 85-90 km, which should result in a negative difference in the clouds in that region, but I see a dipole structure there (possibly only due to clouds being at different altitudes!). In the opposite hemisphere, I don't see any temperature differences, but a big positive signal in the ice volume den-sity and also a (weaker) dipole structure. o P21 L3-4: The negative am-pm difference OF WHAT at 80-85 km at latitudes below (DO YOU MEAN EQUATORWARD?) 80ŮȩN is well anti-correlated to the am-pm ice differences OF WHAT. – Don't understand this sentence. o P21 L4-5: In the NH temperature panel of Figure 12, I see temperature

differences around 0K, are they even statistically significant? Also, shouldn't a positive temperature difference lead to a negative ice volume density difference, but the Vice NH plot shows a positive on? Don't understand this sentence. 24. P22 L3: I wouldn't call Figure 5 showing a "very good agreement" overall 25. P22 L4: slightly larger – please quantify

technical corrections 1. Throughout the text, you mix up the present and the past tense, especially when it comes to reporting results from previous papers. I think that the paper would read better if the present tense was used consistently throughout the paper. Examples: P1, L6, P1 L14, P2 L18-19, P2 L22-35, P3 L90-94, P4 L11 – P5 L8, P5 L16-17, ... 2. You mostly use both terms PMCs and NLCs, while I think it would be more consistent to stick to one term throughout the paper. 3. P1 L1: we have analyzed MIPAS IR measurements ... 4. P1 L3: coverage of the PMC total ice volume 5. P1 L13: caused by sequestration and sublimation 6. P1 L14: latitudes POLEWARD of 70°. Or do you only mean northern latitudes only? Then I would write latitudes poleward of 70°N 7. P1 L15: the PMC volume ice density 8. P1 L17-18: occur IN the coldest region of the atmosphere near the summer POLAR mesopause 9. P2 L7: (e.g., Baumgarten and Fiedler, 2008; 10. P2 L9: (e.g., Berger and Zahn, 2002 11. P2 L12: global climate change in the middle atmosphere 12. P2 L14: Since enhanced CO2 amounts (see, e.g., Yue et al., 2015) ARE EXPECTED TO lead to 13. P2 L14: and higher CH4 amounts to enhanced 14. P2 L15: they MAY both lead to 15. P2 L16: in the MIDDLE atmosphere (middle atmosphere is defined as the stratosphere and mesosphere) 16. P3 L4: A different technique, however, has been DEVELOPED by the AIM/SOFIE (Aeronomy of Ice in the Mesosphere/SolarOccultation for Ice Experiment) instrument. You should also name this different technique in this paragraph. 17. P3 L12-13: write out instrument abbreviations, at least MIPAS! 18. P3 L14: better spatial AND TEMPORAL coverage 19. P3 L16: In A previous paper (López-Puertas et al., 2009) 20. P3 L17: remove MIPAS complete name since you have to put it earlier 21. P3 L20: global distribution 22. P3 L20: of ice volume density 23. P3 L22: and ITS hemispheric dependence ... Since you refer to the layer, not PMCs 24. P3 L22-23:

the correlation of ice volume density 25. P3 L23: and the water vapour concentration 26. P4 L6: during the summer seasons 27. P4 L10-11: described BY López-Puertas et al. (2009). 28. P5 L8: The more recent version 5 (5.02/5.06) of MIPAS L1b spectra has been used – What are the differences to the previous version? Is there a reference for the new version? 29. P5 L12-13: were retrieved only for scans with converged pressure-temperature profiles 30. P5 L18 –P6 L1: are described BY García-Comas et al. (2012). 31. P6 L7: and SHOW, in general, a remarkable agreement 32. P6 L7-8: is there a reference to this statement? 33. P6 L8-10: is there a reference for this statement? 34. P6 L11: Since MIPAS measures PMCs in IR emission 35. P6 L12: particularly ABOUT WHETHER they are warmer or colder 36. P6 L13-14: 5-20 K cooler than the ambient ATMOSPHERE 37. P6 L20: will be heated by absorption 38. P6 L22: warmer than the ambient gas BY about 1 K 39. P6 L26: affected by the problem pointed out by Petelina and Zasetsky (2009) – I think you should briefly describe that problem. If you don't want to do that, please write "affected by A problem pointed out by Petelina and Zasetsky (2009)" 40. P6 L27-28: If we assume that ice particles are cooler than the retrieved gas temperature we obtain 41. P6 L30: The vertical resolution of the ice density profiles 42. P6 L30-31: averaging kernel matrix, depends 43. P6 L32: For the middle and upper atmosphere modes (MA and UA, or together MUA) it is coarser 44. P6/P7: I would combine the information about the vertical resolution and the averaging kernels into one paragraph. 45. Figure 1 caption: Zonal mean ice volume density during four days, two in the SH and two in the NH, . . . 46. Figure 1 caption: is the mesopause as derived from MIPAS 47. Figure 1 caption: explain what #sc means in subplot titles. 48. Figure 2 caption: Polar maps of ice volume density . . . 49. Figure 2 caption: explain what D+N means. Is this information necessary, since it is the same in all four subplots? 50. P8 L5: Note that MIPAS IS sensitive 51. P8 L2 – P9 L5: If you don't comment on the +3K line in the text – why put it in the figure? 52. P8 L6: Noise errors in these plots are about $0.3 \times 10{-14}$ cm3/cm3. – How do you calculate this noise error? 53. P8 L6-7: The PMCs are generally located at regions colder than the frost point temperature for almost all conditions: - Please comment on

the low latitude clouds detected outside regions colder than the frost point temperature: why there? Measurement error, false detections? Due to transport? 54. Figure 1: would it be worth including another contour for 0.3 10-14 cm3/cm3, so one can see which detections may be false clouds? This may be interesting for clouds detected at low latitudes. Also true for many following plots. 55. P8 L10 - P9 L1: Emission from isolated clouds located in the LOS far away from the tangent point, and hence at higher altitudes, can be measured and thus attributed to these lower tangent heights (see ...) – Replace with: Emission from isolated clouds located in the LOS far away from the tangent point is reported at abnormally low tangent altitudes (see ...). 56. P9 L2: the FOV can affect the HEIGHT OF THE lower and upper boundaries 57. P9 L5: see bottom-left panel of Fig. 1 around 70âǬeN. – The bottom left panel shows PMCs in the SH. ? 58. P9 L9: they are sometimes far away – reformulate. 59. Figure 3 caption: Zonal mean ice volume density 60. Figure 3 caption: weighted with the ice volume density 61. Figure 3 caption: for the NLC (top panels) and the MUA (lower panels) modes (since you should have explained what MUA means in the text before this figure caption) 62. Figure 4: It seems you have forgotten to put the SH results as in Fig. 3. Also the ordering is wrong (the two NH plots should be below each other, not next to each other). 63. Figure 4 caption: Zonal mean ice mass density 64. Figure 4 caption: weighted with the ice mass density 65. P10 L6: at slightly lower altitudes in the NH 66. P10 L10-11: because they were taken in the summer on different days, with those of the NLC mode closer to the peak of the PMCs season. Reformulate, e.g., because observations in different modes occurred on different days, with observations in the NLC mode generally occurring closer in time to the peak of the PMC season than observations in the other modes. 67. P10 L14: found in the SOFIE IR extinction measurements - SOFIE also measures scattered radiation at UV wavelengths! See Hervig et al., Interpretation of SOFIE PMC measurements, 2009. 68. P10 L15: measured by lidars - please also add reference for this statement. 69. P10 L15: Hervig et al. (2009b) have shown that 70. P10 L15-16: that the vertical smoothing DUE TO limb view geometry 71. P11 L1: are then more sensitive than lidars to clouds at higher

altitudes. 72. P11 L8: and SOFIE measurements 73. P11 L10-11: at a latitude close to 70° - this is imprecise, especially since you further below state how important the correct latitude is for the bottom altitude. 74. P11 L11-12: SOFIE obtained a slightly lower altitude of 81.6 km, which is within their mutual standard deviations. – In this sentence you write just about SOFIE, so what does "their" and "mutual" refer to? 75. P11 L13-14: whereas in SOFIE those found below 79 km were excluded - reformulate, e.g., whereas SOFIE measurements that had an altitude of peak extinction below 79 km were excluded (Hervig et al., 2009b). 76. P11 L14-15: hence A DIFFERENCE OF a few degrees in latitude 77. P11 L15: might induce 78. P11 L15: in bottom altitude 79. P11 L20: Replace your title "Concentration" with "Ice mass density". I connect the word concentration more to number density, whereas you're writing about the MIPAS ice mass density. 80. P11 L26-27: These SOFIE measurements OCCURRED 81. P11 L27: at latitudes between 66ậŮęN, in the early season, to 74ậŮęN, towards the end of the season - this range does not include any information about the min/max SOFIE latitude at mid-summer. 82. P11 L29: Figure 4 is not introduced properly. 83. P11 L 32-33: that present an average value of 47.4 ngm−3, and the lidar results reported by von Cossart et al. (1999), with ice mass ranging from 36 to 102 ngm−3 – hard to understand, please rewrite. 84. P11 L33 – P12 L1: MIPAS, as well as SOFIE, also measure thinner ice clouds than other IR instruments measuring the PMCs from space, e.g., HALOE (Hervig et al., 2003). – This sentence somehow interrupts the logical flow of comparing MIPAS/SOFIE to lidar, doesn't fit in. Where is this result from, could you refer to specific plots that you used to come up with this conclusion? 85. P12 L6: reported by Hervig et al. (2009b) 86. P12 L7: at least for the 65-75° latitude RANGE 87. P12 L9-10: for the coincident days and latitudes - state your coincidence criterion. 88. P13 L1: large variability of the ice mass density 89. P13 L3: the mean MIPAS ice mass density 90. Figure 5: why do you only show results from the MA and UA modes (MUA), and not the NLC mode? 91. Figure 5: Why do you only show NH data, not SH data? If you don't want to show, at least provide some statement about how they look like. 92. Figure 5 caption: The mean IWC values for both MIPAS and SOFIE are also

provided. (You have already defined IWC in the text, so you don't need to do it here again) 93. P13 L5: (except in 2011) - I don't agree: also in NH2011, the MIPAS ice mass density is higher than the SOFIE ice mass density above 85 km. 94. P13 L5-6: This MIPAS data feature of large ice MASS densities at high altitudes can (remove commas) 95. P13 L6: Fig. 4, bottom right panel – Figure 4 has only one row of plots, there is no bottom right panel 96. P13 L5-9: The whole discussion of high MIPAS ice mass densities at different latitudes does not add to the discussion of Figure 5, which shows a comparison to SOFIE at the SOFIE latitudes. If your point is that MIPAS ice mass density is high compared to other observations at non-SOFIE latitudes, you have to compare to another instrument that observes at more latitudes than SOFIE. 97. P13 L9: In this region – replace with: At the altitudes of the largest high bias in MIPAS ice mass density 98. P13 L15: The mean IWC of the coincident observations, which is provided in Fig. 5, is generally larger . . . 99. P13 L17: in better agreement with model calculations than SOFIE (Hervig et al. (2009c, see their Fig. 5d). 100. P13 L22-23: As expected Figure 6 shows the same general behaviour as discussed above for the volume density (Figure 3). 101. P13 L23-24: I would rewrite this sentence a little: In NLC mode, which contains observations during the mid-season period, we note that the amount of water vapour in the form of ice ranges from 1 to 3 ppmv at latitudes equatorward of 70-75°, and reaches values up to 5-6 ppmv close to the poles. 102. P13 L25: since, to our knowledge, the water ice content has not been measured at latitudes higher than ∼75°. – I don't agree: AIM CIPS measures the IWC at latitudes higher than 75°, see e.g., http://lasp.colorado.edu/aim/browse-images.php. 103. P13 L26: Hervig et al. (2015) have shown time series of SOFIE Qice at the altitude of peak extinction for the 2007-2013 period for the Northern and Southern hemispheres (their Figure 2). 104. P13 L29: shown in the left panel of Figure 6 – do you mean the left panels or the top/bottom left panel? 105. P13 L30: Section 6. 106. P14 L2: Please describe how you define the mean altitude of the PMC layer: e.g., do you use the ice volume density, ice mass density, or Qice? 107. Figure 7: replace "mesopause" and "PMC mean altitude" with zmeso and zPMC, respectively. 108. Figure 7: consider

using a second y-axis on the right side that shows zmeso – zpmc without the 79km off-set. 109. Figure 7 caption: The different colors indicate the results for the NLC (black) and MUA (red) MIPAS observation modes (see Table 2). 110. Figure 8: in the sub-plot titles, remove "Ice water content" since that information is already visible in y-axis annotation. Replace by IWC in y-axis labels and caption. 111. P15 L2: The MIPAS mean valueS obtained here – what are they? 112. P15 L7: is significantly smaller, in ∼1 km - What do you mean here? Did you mean to write: is significantly smaller, by ∼1 km – which would mean that the SOFIE mean PMC height is 2.5 ±0.5 km below the mesopause, OR is significantly smaller, only ∼1km, - which means that the SOFIE mean PMC height is 1 km below the mesopause. 113. P15 L7 - P16 L1: Please add a citation for this statement. 114. Figure 9 caption: Remove "ice water content", use only IWC 115. Figure 9 caption: describe black line and correlation coefficient 116. Figure 9 caption: Zonal mean ice volume density (left) and H2O concentration anomaly . . . 117. Figure 9: if you don't discuss the +3K frost point temperature in the text, why put it into the plots? 118. Figure 9 caption: the mesopause as derived from MIPAS 119. P16 L4: obtained by Russell et al. (2010) 120. P17 L6-7: I would rewrite this sentence as: Figure 8 is consistent with the zonal mean ice volume density shown in Figure 4, which shows that ice mass density increases towards the poles. 121. P17 L7: The large variability is also clearly visible – I would rewrite this as: Large variability from season to season is also clearly visible 122. P17 L11: The NH/SH ratio of the ice water content varies with latitude (not shown) 123. P17 L17: The correlation is significant and shows that the PMC layers are denser and wider when the frost point temperature occurs at lower altitudes. – This sentence needs to be rewritten: how can the PMC layer be denser and wider at the same time? This sounds like a contradiction to me, what exactly do you mean? And why do you talk about (several) PMC layers – do you mean multiple layers in altitude? 124. P17 L23: the SOFIE vertical resolution is well suited 125. P17 L29 – P19 L3: This text interrupts the discussion of Figure 11 and should be moved to the introduction (Section 1). 126. Figure 11: Remove "D+N" from subplot titles 127. Figure 11 caption: note the different scales 128. Figure 11 caption:

**[ACPD](about:blank)**

Interactive
comment

explain what white symbols are 129. Figure 12: It would be useful to use the same x and y axis ranges for all four panels since that makes comparison easier. 130. P19 L4: polar region: the region . . . 131. P20 L1-2: by Hervig et al. (2015) using SOFIE observations 132. P20 L4: observed bottom of the ice layer 133. P20 L8-9: could you add a sentence about how the column abundance of gas-phase $H_2O$ in the dehydration layer and the hydration layer is defined? 134. P20 L15: Diurnal variation of ice volume density 135. P20 L16: The diurnal variation of PMCs 136. P20 L16: when comparing different PMC datasets. 137. P20 L17: equatorward 138. P20 L18: in temperature and meridional advection 139. P20 L24: but not as large 140. P20 L25: 36% in the IWC (not shown). 141. P20 L25-26: Note however that the IWC am-pm differences are also influenced by the slightly negative volume density difference at altitudes above 86 km – Could you explain better how/where from you get the "slightly negative volume density difference at altitudes above 86 km", e.g. point to a plot that shows this? 142. P20 L29-30: That corresponds . . . That indicates . . . - rewrite more elegantly. 143. P20 L30-32: a significantly narrower and thinner pm cloud which mainly disappears below 84 km, in agreement with findings at sub-polar latitudes from Stevens et al. (2010) and Gerding et al. (2013). 144. P21 L4: northward of 65°N. 145. P21 L18-19: We have analyzed MIPAS IR measurements for the NH and SH summer seasons from 2005 to 2012. 146. P21 L21: It is therefore sensitive – What is sensitive? It sounds like scattering is sensitive to the total ice volume. 147. P21 L21: very small ice particles that UV-VIS scattering observations are generally not sensitive to. 148. P21 L23: The measurements cover only a few days of the PMC season (varying from 3 to 15) but, as opposed to SOFIE, have global polar coverage. 149. P21 L30: from lidar measurements 150. P21 L31: particles in the upper part 151. P22 L1: PMCs are very variable 152. P22 L2: concentration – do you mean ice mass density? 153. P22 L2: water ice content – replace by IWC 154. P22 L4-5: smaller sensitivity – do you mean that MIPAS is less sensitive than SOFIE? Do you mean that the "smaller sensitivity" cause slightly larger MIPAS IWC, or a larger MIPAS variability? Please clarify. 155. P22 L5-6: larger values – of what? 156. P22 L15: It is not clear what "their" refers to - please

rewrite more clearly. 157. P22 L15: since the NLC-mode measurements 158. P22 L16: while the MUA-mode observations are 159. P22 L17: as we move away from the polar region – replace with "towards more equatorward latitudes" 160. P22 L18: equatorward 161. P22 L20: from several satellite instruments 162. P22 L25: denser and wider – what do you mean? 163. P22 L27: of MIPAS PMCs and water vapour have shown 164. P22 L27: have confirmed that PMC layers 165. P22 L32: has shown

---

## Author Comment (AC1) · 29 Apr 2016

**Response to the comments of Reviewer #1**

*We are very much indebted and thankful to the reviewer for the very detailed reading of the manuscript and the very helpful comments and suggestions. Although the comments from the reviewer were rated as "minor", they were very helpful for revising the paper. After consideration of all of them a much-improved paper has resulted. Detailed responses (in italics) follow after each comment.*

**General comments**

• **Reviewer's comment:**
General comments: overall quality of the discussion paper This study introduces and describes a PMC dataset from MIPAS IR emission observations, shortly presents the retrieval method that was already published in a previous paper, provides evaluation of its quality by comparison of MIPAS ice mass density and cloud altitude to AIM SOFIE observations, and discusses MIPAS cloud properties in relation to previous findings. The important advantages of an IR emission dataset like MIPAS compared to other (UV and/or VIS) remote sensing datasets are: o that observations are available during day and night o but also that retrieved cloud properties are independent of the highly uncertain PMC particle size distribution, which has to be assumed for the retrieval of cloud properties of many other satellite datasets. As such, this paper represent a substantial contribution in the form of new data to scientific progress within the scope of Atmospheric Chemistry and Physics. While the paper would profit from proof-reading by a native English speaker, there are also weaknesses in the discussion of the MIPAS measurement threshold, the discussion of the disagreement with SOFIE ice mass density profiles above 84km (Figure 5), and the discussion of MIPAS diurnal variations. A more thorough evaluation of the dataset could be achieved by additional comparison to another PMC dataset that also offer polar coverage, however, that would probably lengthen this paper too much. On the other hand, the paper can be shortened by emitting results that are not discussed in detail, for example Section 5. The conclusion should be more quantitative when summarizing the agreement with SOFIE. In summary, I think that all the here mentioned weaknesses can be resolved using the existing dataset, so I recommend this paper for publication in ACP with minor revisions.

**Responses:**
*We are glad to hear that this work represents a "substantial contribution".*

*There was already available a new version revised by a native English speaker where many of the suggested corrections were already done. This was probably caused by the kind of double-review system of ACP.*

*Figure 5 has been revised, following the suggestion of the other Reviewer that we should considered the solar local time when comparing MIPAS and SOFIE observations. As a result the agreement is now much better. The text discussing this comparison and the Conclusions section have been revised (see comments below).*
*Also, the section on the diurnal variation has been significantly revised including the figures (Figs. 10) (see comments below).*

*We agree that comparison with other instruments should be done, specifically with the IWC of CIPS. However, as recognized by the reviewer, this would lengthen the paper too much.*

*About the shortening of Sec. 5, the other reviewer actually suggested to discuss the correlation separately for each hemisphere. This section is already short and reports a finding that we consider important. Hence we have kept the text (actually expanded as suggested by the other reviewer) but it has been merged with the previous section. Figure 9 has also been reduced to just one panel.*

*We have revised the conclusions about the MIPAS-SOFIE comparison being now more quantitative.*

**Specific comments: individual scientific questions/issues**

• **Reviewer's comment:**
1. You compare the MIPAS PMC dataset to SOFIE, which observes PMCs at just one latitude each day. This latitude is slowly varying during the PMC season, but basically this restricts your comparison to a narrow latitude range. Have you considered comparing your dataset to other satellite observations, e.g., from CIPS, OSIRIS, SCIAMACHY, ...?

**Response:**
*There are two major reasons of why not extending the comparison of MIPAS data to other instruments. First, the difficulty in choosing a common quantity that characterizes the PMCs. As discussed, MIPAS measures the ice volume density, irrespective of their particle size distribution. Most instruments measuring in the UV-VIS are not sensitive to the small ice particles existing in the upper part of the PMC layers; thus they are not the most appropriate instruments for comparison. CIPS might be also useful but provide, to our knowledge, only the integrated column ice (IWC), not profiles. BTW, we already considered the comparison with CIPS a couple of years ago, in conversations with Cora Randall, but comparing cloud coverage is difficult and certainly not very quantitative. Anyway, as you say, that comparison would lengthen the paper too much.*

• **Reviewer's comment:**
2. You retrieve the ice volume density as it is independent of the assumption of the particle size distribution, which is considered uncertain. Have you considered retrieving the particle size and number density, using the same assumption that the SOFIE team uses?

**Response:**
*MIPAS does not have information on the particle size. SOFIE extract that information using different channels at different wavelengths in absorption. Unfortunately this is not possible with MIPAS. This is a limitation of the **IR emission** technique. We are sensitive to the volume emission but cannot distinguish the particles' size.*

• **Reviewer's comment:**
3. Do you have plans for making this dataset publicly available?

**Response:**
*Yes, we participate in MesosphEO, a project of ESA, and it is planned that we provide the PMC dataset as part of our duties.*

- **Reviewer's comment:**

  4. Your introduction should state clearly which SOFIE version you are using. I have found this information in the figure caption of Figure 5, but it belongs in the introduction.

  **Response:**

  *We agree. We have now included in the introduction that we use version 1.3 of SOFIE data.*

- **Reviewer's comment:**

  5. P3 L24-25, also P8 L5: I'm missing a more careful discussion of the MIPAS measurement threshold as done by Hervig et al., Interpretation of SOFIE PMC measurements . . . (2009). Some points regarding the SOFIE detection threshold and how it affects the radius retrieval from that paper: - The SOFIE ice detection threshold corresponds to Mice~0.06 ngm3. - It is important to note that particle size is only determined when the extinction β(1.037) is above the noise. - Although SOFIE can rarely determine particle size for the most tenuous clouds, size is characterized over the dominant range of measurements.

  **Response:**

  *As an IR emission instrument, the detection limit is usually imposed by the instrument's noise mapped into the retrieved quantity, the ice volume density in our case, in the retrieval scheme. We have indicated this value in the single measurements as well as when doing any kind of averaging, which reduces it by the square root of the number of averaged data.*
  *Since we do not retrieve the particles' size, we have no detection threshold in the sense discussed in SOFIE measurements.*

- **Reviewer's comment:**

  6. P3 L30: It also operated with a high sensitivity – How do you define high sensitivity?

  **Response:**

  *The noise equivalent spectral radiance (NESR) of the 5 MIPAS bands are detailed in the given reference of Fischer et al. (2009). In particular, in the spectral region were the measurements analysed were taken (shorter-wavelength end of band A), is about 20 nW/(cm$^2$ sr cm$^{-1}$) (it slightly changes from spectrum to spectrum and with wavelength). This was already stated in L1 of page 5. The reference was also given already.*

- **Reviewer's comment:**

  7. P6 L8-10: Please comment on how a potential 5-10K cold or warm bias affects your retrieved clouds. Do you believe The MIPAS temperature measurements more/less than those of other instruments?

  **Response:**

  *This point is discussed in more detail and quantitatively in P7, lines 4-7. A 5 K error in temperature induces approximately (Planck function at 145 K and 800 cm$^{-1}$) an error of about 30%.*
  *About your second question, it is difficult to answer. The detailed validation carried out by García-Comas et al. (2014) have shown that MIPAS temperatures are in between SABER, MLS and OSIRIS on one side, and ACE and SOFIE on the other. One major point here is the vertical resolution, rather good for SABER and ACE, moderate in the case of MIPAS and rather coarse for MLS. SABER, however, as well as MIPAS, are prone to non-LTE effects and to*

*uncertainties in the atomic oxygen, while ACE is free of non-LTE. We are very confident in our retrieved MIPAS temperature but certainly cannot discard a systematic error of +/- 5 K near the polar summer mesopause.*

- **Reviewer's comment:**
  8. Regarding all figures plotted vs. altitude: shouldn't the vertical axis be tangent altitude, not altitude?

  **Response:**
  *No, it shouldn't. We normally use tangent height if we plot the measured **radiance** at the limb tangent heights (see, e.g., Fig. 4 in López-Puertas et al., 2009). However, after we perform the retrieval, the retrieved quantities (temperature, H2O vmr and ice volume density) are all expressed in actual vertical heights in the atmosphere. See, for example, how radiances in the mentioned figure, due to the limb geometry are very large even below the PMC layer. Note this is not the case in the quantities shown in this paper.*

- **Reviewer's comment:**
  9. Do you have enough lines of sight through one cloud volume that you could apply a tomographic algorithm? Such an algorithm has the advantage that it solves the problem of clouds in the fore- or back- ground being assigned anomalously low tangent altitudes. You mention this problem on P8/9. Due to this problem SOFIE discards all clouds below a tangent altitude of 79 km. Examples for tomographic algorithms applied to PMC data: Hultgren et al., First simultaneous retrievals of horizontal and vertical structures of Polar Mesospheric Clouds from Odin/OSIRIS tomography, 2013; Hultgren and Gumbel, Tomographic and spectral views on the lifecycle of polar mesospheric clouds from Odin/OSIRIS, 2014.

  **Response:**
  *We did not try but it would worth to do, at least on an orbit-by-orbit basis. However, the current version of our retrieval processor does not have this capability. Extending the processor for including this option would take rather long, certainly beyond the deadline for submitting these responses. This is, however, a very point that we will consider for the future improvements of the retrieval.*

- **Reviewer's comment:**
  10. P9 L8-9: Please comment on why in the upper left panel of Figure 2, the clouds fill only a small portion of the region where the temperature is below the frost point temperature, while on the lower left panel (also SH) the cloud coverage seems to agree much better with the frost point temperature boundary.

  **Response:**
  *The fact of the temperature being smaller that the frost temperature for having PMCs is a necessary conditions but not sufficient. Other factors that influence the presence of PMCs are availability of nuclei for condensation, sedimentation, transport, ice growth and sublimation time dependence, and particle size effects on saturation vapor pressure. On the other hand, even assuming thermodynamic equilibrium, T_frost is the temperature below which ice formation is possible but only at 5-10K lower temperatures the ice growth really becomes asymptotic (Hervig et al., 2009). That is, we do not expect having ice clouds everywhere*

*temperature is below the frost temperature. Finally, we estimated T_frost from a constant mean water vapor profile, whereas water vapor changes. This variability has not been taken into account in the evaluation of our T_frost.*

- **Reviewer's comment:**

  11. P9 L8-9: Please comment about the possibility that you see the effect of 5-day planetary wave activity in Figure 2. You could also comment about the possibility to use your dataset to help track the effect of space shuttle exhaust on PMCs, e.g., Stevens et al., Antarctic mesospheric clouds formed from space shuttle exhaust, 2005; Stevens et al., Bright polar mesospheric clouds formed by main engine exhaust from the space shuttle's final launch, 2012; Stevens et al., Polar mesospheric clouds formed from space shuttle exhaust, 2003.

  **Response:**
  *MIPAS was observing in the middle/upper atmospheric mode during and right after the launch of the space shuttle in July 2011 (see Table 1). We have quickly looked at that and found hints about increasing of the ice density near ALOMAR on 9th July 2011. However, this requires a more careful analysis that is beyond this work.*

- **Reviewer's comment:**

  12. P11 L12-13: reformulate sentence, e.g.: Using the NLC mode data at similar NH latitudes, we derive a mean bottom altitude of ~81 km, slightly lower than that of SOFIE. But I think "similar NH latitudes" is too imprecise. It is not clear how exactly you choose the MIPAS latitude for comparison to SOFIE: do you use one mean latitude value, or a daily changing latitude value based on the changing SOFIE latitudes? Please describe your method better.

  **Response:**
  *We now write that the comparison is done with MIPAS zonal means in a latitudinal band extending ±2 deg around mean latitude of SOFIE's measurements.*

- **Reviewer's comment:**

  13. P11 L13-14: Note, however, that we have not excluded any PMCs here, whereas in SOFIE those found below 79 km were excluded. – You're not comparing apples and apples: what happens when you treat MIPAS observations like SOFIE did? Do you then get a better agreement (as expected)?

  **Response:**
  *When treating MIPAS observations like SOFIE the change is very marginal, only 0.1 km. We have re-written the sentence to:*
  *"In SOFIE measurements the PMCs with a peak extinction altitude below 79 km were excluded (Hervig et al., 2009b). Applying a similar threshold to MIPAS data, however, does not change significantly the bottom altitude.*

- **Reviewer's comment:**

  14. P11 L28: at those latitudes – please be more precise: what is your coincidence criterion? It may be worth showing a histogram of SOFIE and MIPAS ice mass. This would be helpful in convincing the reader of the nice agreement.

  **Response:**

*The coincidence criterion has now been specified, within 2 degrees of SOFIE latitude measurements.*

- **Reviewer's comment:**

  15. P12 L1-3: I don't understand your explanation why MIPAS and SOFIE are expected to observe less ice mass density than the lidar: if MIPAS and SOFIE are able to observe a BIGGER population of the total ice mass by ALSO observing the smaller particles (that the lidar does not observe), shouldn't the resulting ice mass density be BIGGER than the lidar ice mass density?

  **Response:**
  *Our reasoning (see also Hervig et al.) is that if one instrument samples only the larger clouds, it is clear that one would get a mean bias to larger values with respect to another that sample all values, large and small.*

- **Reviewer's comment:**

  16. P13 L9-10: ice particles are the smallest and it could be that MIPAS is more sensitive than SOFIE to those particles. – Here you argue that a more sensitive instrument should result in higher values of ice mass density. On the previous page you have argued the opposite: that the larger sensitivity of MIPAS to the smaller ice particles than lidar will lead to lower mean ice mass density values. It makes the impression as if you are contradicting yourself.

  **Response:**
  *We agree that it looks contradictory.*
  *We have removed that sentence.*

- **Reviewer's comment:**

  17. If you haven't done that yet, I would suggest talking to Mark Hervig directly about possible reasons for the disagreement in ice mass density above 84km as seen in Figure 5. Please also discuss possible reasons for this disagreement in more detail, e.g., the role of geophysical differences.

  **Response:**
  *Thank you very much for the suggestion. We did not have the opportunity to talk to Mark Hervig about these differences. The differences in the revised version, however, are much smaller. As suggested by the other reviewer, we have now considered the criterion of having the closest possible solar local time. This has greatly improved the comparison.*

- **Reviewer's comment:**

  18. P17 L15-17: The discussion of Figure 9 is very short (2 sentences) and contains only a description of Figure 9. Do you have an overall point you want to get across with Figure 9? Is this a new result or do you show this to relate MIPAS observations to previous studies (which?)? Otherwise, please consider omitting Figure 9 and Section 5.

  **Response:**
  *It is true that the discussion is short and may do not deserve a Section on its own. We have merged it with the previous section.*

*However, the result is clear, we think. The major point being that when the atmospheric temperature is below the frost point temperatures at lower altitudes, the PMCs are dense (IWC is larger). We think this point is already clear in the text. To our best knowledge, we are not aware of any previous study on this.*

- **Reviewer's comment:**
  19. Last paragraph of Section 6: the discussion about column abundance of gas phase H2O around 70◦ is not supported well by Figure 10, which shows the gas-phase H2O vs. altitude and latitude.

  **Response:**
  *That is correct. There are several points here. First, around 70 deg, the distinction between the hydration and dehydration layers in MIPAS data is not so clear as reported by Hervig et al. However it is evident at higher latitudes.*
  *Secondly, the quoted values of MIPAS H2O anomalies in both regions should be given as integrated columns and not as the peak values in both regions. When integrating, the column in the hydration region, 6-7 ppmv*km, is about twice larger than in the dehydration region, 3.5-4.5 ppmv*km. On this point MIPAS and SOFIE do not agree well.*
  *Third, these anomalies in the gas-phase water are both much smaller than the amount in ice. On this point both SOFIE and MIPAS agree very well.*

  *We should also have in mind that in order to calculate accurately the excess and deficit of water vapour in the lower and upper regions, we should use the H2O gas profile corresponding to the same geolocations but with no ice. Such profile (the background profile of Hervig et al.) was estimated by Hervig et al. from measurements where no ice was present. To determine such profile in the case of MIPAS is very difficult because of the high noise in the single profiles of the retrieved water vapour. Hence, we used as the "no-ice" H2O gas profile the zonal mean profile corresponding to all latitudes. This could partially explain the SOIFE and MIAPS differences.*

  *The text has been revised along these lines.*

- **Reviewer's comment:**
  20. P20 L1: exhibit a very good latitude/longitude spatial correlation – I don't agree: while the dehydrated "hole" in H2O at 90 is neatly centered on the pole, the clouds' center of mass is shifted towards northern Greenland, and at 80km the center of mass of the hydrated region is over the northern Pacific. I wouldn't call this "very good latitude/longitude spatial correlation", but expect a comment on this "rotation".

  **Response:**
  *The comment is very pertinent. We agree that we should not say a "very good ... correlation". It is the first order broad feature what we want to highlight. Also, we do not expect a perfect correlation, as is not expected an immediate response neither the same structure at different altitudes (e.g. propagation of gravity and planetary waves). The text has been changed in this sense.*

- **Reviewer's comment:**
  21. P20 L4-5: I don't agree with your statement that the location of the hydration region agrees well with SOFIE observations. From Figure 10 it looks like the MIPAS peak altitude

of the hydration region is at 80km, whereas the bottom of the PMC layer is at 81 km. If anything, then the MIPAS peak in hydration lies BELOW the bottom of the PMC layer. Or do you also count the dark blue shading as PMC? Then I would agree. But for that it would be useful to know if these dark blue PMC observations are above the noise threshold.

**Response:**
*You are fully right. We understood that the peak altitude of the hydration region in SOFIE is 0.3 km BELOW the bottom PMC layer but it is ABOVE.*
*We have corrected the text stating now that they do not agree.*

• **Reviewer's comment:**
22. P20 L9-10: What do you mean with being "more pronounced"? That the ice layer contains even less H2O than the hydration/dehydration layers? At higher latitudes the dehydration (-0.3 or -0.4 ppmv maybe) looks much less pronounced than the hydration (1.4 ppmv), which does not agree with the SOFIE results that they are roughly equal. But again: my misunderstandings could be solved by showing a plot of the H2O column abundances.

**Response:**
*Sorry for the misunderstanding. We meant that the excess/deficit of of H2O gas phase in MIPAS data are both larger (in absolute values) at latitudes closer to the pole; i.e., they increase from 70ºN towards the pole.*
*We did not include extra figures but calculate the columns and included the values in the text. See also the response above.*

*The text has been revised along these lines.*

• **Reviewer's comment:**
23. Section 7 (Diurnal variation of ice volume density, Figure 12) is lacking clarity and not convincing:

**Response:**
*We have re-written the whole section and corrected several typos. We think that the section reads now more easily.*

o P20 L26-28: as you write, there is an altitude difference between the morning/evening clouds, and you note that this altitude difference leads to the altitude bipole structure in Figure 12. Is it possible to correct for the altitude difference in order to get rid of the bipole structure?

*The alternating negative-positive differences are actually indicative of a change in the mean cloud altitude with the NH pm clouds being on average at lower altitudes at 65-75N. Additionally, the shape of the average Vice is not the same during am and pm. Therefore, a simple vertical displacement of the am clouds would not make the bipole structure to disappear.*

o P21 L1-2: These ice volume density differences are remarkably anti-correlated with the 10 am-10 pm differences in the kinetic temperature measured by MIPAS – I don't agree: there is a positive difference in T at 60-70S and 85-90 km, which should result in a negative

difference in the clouds in that region, but I see a dipole structure there (possibly only due to clouds being at different altitudes!). In the opposite hemisphere, I don't see any temperature differences, but a big positive signal in the ice volume density and also a (weaker) dipole structure.

*We have now zoomed-in Fig. 12 in order to clarify the discussion. We have also refined the discussion. We write now that the good anti-correlation is only found in the NH. That is more clearly seen in the relative $V_{ice}$ differences, that we now include in the manuscript. We also more clearly state that the SH ice differences are not well anti-correlated with temperature.*

o P21 L3-4: The negative am-pm difference OF WHAT at 80-85 km at latitudes below (DO YOU MEAN EQUATORWARD?) 80ºN is well anti-correlated to the am-pm ice differences OF WHAT. – Don't understand this sentence.

*We re-structured and completed the sentence: 'The positive am-pm ice differences at 80-85 km equatorward of 80N are anti-correlated to negative am-pm temperature differences.*

o P21 L4-5: In the NH temperature panel of Figure 12, I see temperature differences around 0K, are they even statistically significant? Also, shouldn't a positive temperature difference lead to a negative ice volume density difference, but the Vice NH plot shows a positive on? Don't understand this sentence.

*As written in the previous version, there is a tendency of positive temperature differences that is reflected in negative ice differences. We think that it is more clearly seen in the new zoomed-in Fig. 12.*

• **Reviewer's comment:**
24. P22 L3: I wouldn't call Figure 5 showing a "very good agreement" overall

**Response:**
*Even if the new comparison (Fig. 5) is much better that before, we agree, we should not say "very good agreement" overall. Change to "good agreement".*

• **Reviewer's comment:**
25. P22 L4: slightly larger – please quantify

**Response:**
*With the new comparison the differences in IWC are small, ~10%.*

**Technical corrections:**

*We thank again to the Reviewer for all the technical corrections that resulted in an improved manuscript. They have been all been included except a few exceptions as explained below. We also response below to the questions risen in this Section.*

2. You mostly use both terms PMCs and NLCs, while I think it would be more consistent to stick to one term throughout the paper.

*We mainly used the term PMC. We only use "NLC" once in the introduction, to say that PMC and NLC are the same phenomena. To avoid misunderstanding, we have deleted 'as seen from the ground'. We use NLC in many cases when referring to the MIPAS "NLC-mode" measurements. This "NLC-mode" was defined in the MIPAS mission plan as a particular observation mode and hence we have kept its name.*

P8 L6: Noise errors in these plots are about 0.3 × 10−14 cm3/cm3. – How do you calculate this noise error?

*By the standard error of the mean, i.e., by dividing the single noise error by the square root of the number of averaged profiles.*
*We state this now explicitly in all figure captions and in the text.*

Please comment on the low latitude clouds detected outside regions colder than the frost point temperature: why there?

*We include the following text: "Weak PMCs located at latitudes equatorwards of about 60 degrees and outside of the frost point temperature contour are likely false detections caused by instrumental (most likely offset) errors."*

62. Figure 4: It seems you have forgotten to put the SH results as in Fig. 3. Also the ordering is wrong (the two NH plots should be below each other, not next to each other).

*Correct. The caption was wrong. It has been corrected in the revised version.*

Figure 5: why do you only show results from the MA and UA modes (MUA), and not the NLC mode?

*Because we did not want to mix up measurements with different vertical resolution and those of MUA have a better statistics.*

93. P13 L5: (except in 2011) - I don't agree: also in NH2011, the MIPAS ice mass density is higher than the SOFIE ice mass density above 85 km.

*This has changed in the new figures (using only MIPAS pm data). The agreement is better now.*

P13 L25: since, to our knowledge, the water ice content has not been measured at latitudes higher than ←⋯75. – I don't agree: AIM CIPS measures the IWC at latitudes higher than 75, see e.g., http://lasp.colorado.edu/aim/browse-images.php.

*Thank you. The paragraph has been removed. Has this been reported in published papers?*

P17 L29 – P19 L3: This text interrupts the discussion of Figure 11 and should be moved to the introduction (Section 1).

*We agree that, as it is written, the H2O retrieval description interrupts the discussion. However, it is too short and would also be isolated if moved to the "MIPAS measurements" section, which is mainly devoted to ice volume density retrieval. We have re-arranged the text in Section 6 so it does not interrupt the discussion that much now.*

---

## Author Comment (AC2) · 29 Apr 2016

**Response to the comments of Reviewer #2**

*We are very much grateful to the reviewer for the very helpful comments and suggestions. After consideration of all of them a much-improved paper has resulted. Detailed responses to the Reviewer (in italics) follow after each comment.*

- **Reviewer's comment:**
Major Comments:
Given the limited amount of data used from MIPAS, it would be useful to better quantify the observed variability in PMC properties throughout the paper. For example, what is the range of "top altitudes" from MIPAS? If there is a high degree of variability, the mean calculated from 12 or 19 days is likely insufficient to converge on the "true" mean.

**Response:**
*We agree with the referee on this point. We stated already in several instances of the manuscript the large variability of PMCs in MIPAS measurements, caused not only by their natural variability but also for the large single measurement noise (already included in the text). The variability is also evident in the daily zonal mean examples shown in Fig. 1. It is true, however, that we did not provide a quantification of the variability in some cases (note that in most of the zonal mean figures and maps we already stated the estimated noise error). We have quantified the variability of the "top altitude" and found that it is rather large (1 sigma values changing from 1.6 to 2.7 km, depending on the mode of measurement and latitude).*

***The next couple of sentences have been added in Sec. 3.1:***
*"The highest altitude of PMCs derived from MIPAS NLC mode measurements is highly variable, as can be seen in the typical examples shown in Fig. 1. At 70N, it is about 88.5 km (Fig. 3b). Its variability depends on latitude and takes 1-sigma values from 2.7 km near 70º to 1.6 km near the pole."*

*We have also estimated the variability of the bottom altitude and of the mean altitude layers (see responses below).*

- **Reviewer's comment:**
I would check Figure 5, as I would expect better agreement with SOFIE based on the rest of your analysis. In 2009, for example, the average of 175 MIPAS profiles at 87 km is ~15ng/m^2, compared to ~2ng/m^2 from 165 profiles from SOFIE. If this figure is correct, then I think the large differences and variability compared with SOFIE call into question the entire analysis. It also looks like you may even be seeing ice above 90 km in Figure 5 and are simply setting these values to 0. Also, Figure 4 and Figure 5 do not seem to be in agreement. 2009 and 2010 show a peak in mass density above 87 km that is not represented at all in Figure 4.

**Response:**
*This is a very good point. We have revised the comparison following the reviewer's comment below suggesting that we should compare MIPAS measurements with a similar local time as those taken by SOFIE. Thus, we have re-done Fig. 5 including ONLY 10 pm measurements, which are closer to SOFIE's local time of observations in the NH. The agreement is much*

*better now (see new Fig. 5 as Fig. 1 below) because of the smaller ice concentration at 10 pm, particularly at altitudes of 85-87 km (see Fig. 12). The agreement is better now for all years in the region above 85 km. Still we obtain a significantly larger ice amount above around 85 km for 2009 and 2011, although, they are of similar magnitude than the peak below. It is also worth noting that MIPAS does not have a vertical resolution as good as SOFIE. However, the column amount is a more comparable quantity. We see that even for these years, the agreement in the column amount is very good 46 μg/m2 (SOFIE) vs. 41 (MIPAS) for 2009, and 60 μg/m2 vs. 56 for 2011. Thus, although the noise in MIPAS measurements is rather large and the vertical resolution is not as good as in SOFIE, we do think that our measurements are very valid and the analysis is trustworthy.*

**Action: Fig. 5 has been revised and the text has been accordingly revised.**

- **Reviewer's comment:**
  It also looks like you may even be seeing ice above 90 km in Figure 5 and are simply setting these values to 0. Also, Figure 4 and Figure 5 do not seem to be in agreement. 2009 and 2010 show a peak in mass density above 87 km that is not represented at all in Figure 4.

  **Response:**
  *That is correct. In some days there appear some small radiances at tangent heights above 90 km which appear as ice in the retrieval. We think this is not real but caused by the large variability in the offset of MIPAS spectra, which, when integrating over the large spectral range of 770–920 cm$^{-1}$ (see Lopez-Puertas et al., 2009) results in some significant radiance-integrated offset. This offset changes significantly with latitude and season and also with altitude. We have improved significantly the offset correction from Lopez-Puertas et al. (2009) (see the text) but there are still some scans that show some signal above 90 km. Since we think there is no physical reason for attributing this signal to PMC's we did not show it.*

  *About the possible disagreement between Figs. 4 and 5, it is apparent. Fig. 4b shows the mean of all the years, not only 2009 and 2010 but also 2008 and 2011. The latter show also a peak (more pronounced in 2011) at lower altitudes. As a result, when averaging over all years, it results in a kind of broad peak near 70º N extending from 83 km up to 87 km, as shown in Fig. 4b. Actually, we do see a very small increase in this broad peak at about 85-87 km (follow the "10" contour line in Fig. 4b).*

- **Reviewer's comment:**
  Comments:
  General: make sure you define each acronym once, the first time it appears.

  **Response:**
  *We have revised the acronyms and have deleted a few double definitions. We have retained, however, the duplicity between the abstract and the rest of the text and also when they appear in the figure caption (in order to facilitate the reading). Also, we have duplicated the definition of uncommon terms like MA and UA, when they appear in the text far away from where they were defined. Again, we think the addition of these few extra words justify the much easier reading of the paper.*

- **Reviewer's comment:**

Line 25: can't temperatures be lower that 150K?

**Response:**
*Yes, they can. We have replaced "as low as " by "about".*

- **Reviewer's comment:**
  The paragraph beginning at line 35 provides little information except to say that PMCs have been studied. What did these papers show?

  **Response:**
  *The idea with this sentence was to express that PMCs have been measured by many instruments using different techniques and also from the modeling point of view. This is a kind of summary of major observations and modeling efforts. The particular aspects derived from some of these instruments are detailed in the following paragraphs.*

- **Reviewer's comment:**
  Line 55: This paragraph could be reduced to say that similar results were found by Stevens and Hervig [2014] using SBUV.

  **Response:**
  *The paragraph actually contains two pieces of information, one about the temporal evolution of PMCs and another one about the SBUV and SOFIE data comparison. We have re-written the paragraph to:*

  *Similar results were found by Hervig and Stevens [2014] by using SBUV data and a different method for calculating the ice water content (IWC). These authors also compared SBUV and SOFIE data and found good agreement in average IWC if an appropriate threshold was applied to the SOFIE data set and consistent day-to-day and year-to-year variations between both data sets were used.*

- **Reviewer's comment:**
  Line 70: What do you mean by "the responses"? Are you saying that the 27-day solar cycle somehow accounts for long term PMC trends?

  **Response:**
  *No, we meant to say that the 27-day solar cycle variations were not induced directly by the 27-day solar cycle variations in the solar flux but by 27-day variations in the vertical winds. The paragraph has been written to:*

  *Thomas (2015) have studied the solar-induced 27-day variations in polar mesospheric clouds using 15 seasons of SOFIE data and suggested that the 27-day variations in the PMCs are due to 27-day variations of vertical winds.*

- **Reviewer's comment:**
  Titles of Figure 4 are not consistent with figure caption. It looks like the figure caption is wrong.

  **Response:**
  *Correct. Sorry about that. It was a leftover of a previous figure.*

- **Reviewer's comment:**

  In Figure 5, put the red line on top of the shading.

  **Response:**

  *Done.*

- **Reviewer's comment:**

  What is the reasoning for showing a single day in Figures 10 and 11? Wouldn't it make more sense to do this analysis using all the data, so you could more easily compare with previous work?

  **Response:**

  *There are several reasons. First, not all the H2O middle and upper atmosphere data have been processed but only the period of the NLC season in 2005. Secondly, it was not the main aim of this paper to present a detailed study of the analysis of the PMCs (ice) and gas phase H2O data. This is beyond the scope of the paper. However, we thought it is useful to illustrate this as an example since very few (if any) instruments are able to measure global-latitude fields of temperature, ice, and water vapour simultaneously. A more quantitative and extended study is planned for the future.*

- **Reviewer's comment:**

  Why do you show latitudes equatorward of 50◦ in Figure 12? Also, the temperature anomalies do not seem to correspond to the anomalies in ice volume density. Maybe this is because you are comparing January and July differences in temperature to full season differences in ice volume density.

  **Response:**

  *The main reason for showing latitudes equatorward of 50º was to illustrate global tidal features and that the temperatures differences at mid-lat. and near the polar region are due to tides. We agree that it is much easier for looking at correlation between ice concentration and temperature to show the same altitude/latitude coverage in both figures. Thus, this figure has been re-plotted along this line. Note also that the temperature differences have been plotted for the same days of measurements of the PMCs so they are directly comparable now. We have also included new panels with the relative am-pm ice volume densities differences and removed a few outliers (with $V_{ice}$>4-sigma).*
  *In the revised version of the text we have also refined the discussion on the temperature and the ice anti-correlation and state that it is not clear in the SH.*

- **Reviewer's comment:**

  Also, how do your results in Figures 12a and 12b affect your comparison to SOFIE in Figure 5? Would it make sense to only compare am or pm to SOFIE? SOFIE observes sunrise in the NH summer and sunset in the SH summer.

  **Response:**

  *This is a very good point. As described above we have now compared MIPAS pm with SOFIE NH measurements, taken around 23pm on average, (revised Fig. 5) and the agreement is much better. Thank you very much for pointing this to us.*

- **Reviewer's comment:**
  Figure 9 seems to show two distinct populations for the NH and SH. I don't think it makes sense to do a regression analysis of both hemispheres. Looking at the left panel, it seems that there is a strong linear trend in the NH, but in the SH, ice water content seems independent of frost point altitude. Maybe expand your analysis to discuss hemispheric differences and compute the correlation in each hemisphere separately.

  **Response:**
  *If it is a physical reason for such a correlation it is not clear why it should happen in one hemisphere and not in the other. For that reason we presented the analysis together. It is true that for the NLC mode (left panel) there is a stronger correlation in the SH than in the NH (note that this is contrary to what you state). However, the correlations for NH (black) and SH (red) are similar for the MA+UA modes (right panel).*
  *We have now repeated the analysis for each hemisphere separately and found the same conclusion (see figures 2 and 3 below). We find overall a rather good correlation except in the case of the NLC mode in the NH. This could be caused by the smaller statistics we have for this case.*
  *Thus, because we have found the same behavior separately in each hemisphere, and the other reviewer suggested to shorten/remove this section, we have replaced the previous 2-panel figure by one panel showing the analysis for all modes + the two hemispheres together.*

- **Reviewer's comment:**
  Line 217: There is no 70∘N in the bottom-left panel of Figure 1

  **Response:**
  *Correct. It refers to a previous figure showing the zonal mean distribution for the 5th July 2009 in the NLC mode. We have removed this reference in the revised manuscript.*

- **Reviewer's comment:**
  Figure 2: Any thoughts on what drives the zonal variability observed here (i.e., planetary waves such as the 2-day or 5-day wave)? See Siskind, Nielsen, and Merkel

  **Response:**
  *Yes, it could be driven by PW activity. However, MIPAS was operating in the middle/upper atmospheric modes only occasionally (see Table 1) and unfortunately the time sampling does not allow for unambiguous determination of individual periods of zonal oscillations. Effects of the 2-day and 5-day waves are aliased with stationary waves and tides and cannot be isolated. Nevertheless, we now mention the possibility of the wavenumber-1 longitudinal variations in the plot being due to planetary waves, but mainly based on findings from other authors.*

- **Reviewer's comment:**
  Line 261: What is the standard deviation for MIPAS?

  **Response:**
  *The standard deviation of the bottom altitude of PMCs in MIPAS varies with the mode of observation from +/-1.2 km for the NLC mode to +/-1.8 km for the MUA mode. Thus, the bottom altitude in MIPAS is 80.9+/-1.2 km for the NLC mode and 80.0+/-1.8 km for the MUA mode. These data have been included in the text.*

- **Reviewer's comment:**
  You talk about Figure 4 like it is a 4 panel plot, but it only has 2 panels

  **Response:**
  *Correct. Sorry about that, it is a leftover of a previous version of the figure. The text has been changed: "Fig. 4b and 4d" to "Fig. 4a and 4b", and the legend of the figure appropriately corrected.*

- **Reviewer's comment:**
  Line 295: Bardeen et al. [2010] has done this exact analysis.

  **Response:**
  *I think this is a misunderstanding. We refer to a detailed comparison between CARMA and **MIPAS data**, covering high latitudes, not with SOFIE data. We have clarified this in the text: "A thorough comparison with the CARMA model and MIPAS data, including higher latitude regions, would be very useful but is beyond the scope of this paper."*

- **Reviewer's comment:**
  Line 355: What lidar measurements, and at what latitude?

  **Response:**
  *With ALOMAR lidar measurements in Norway (69ºN). This has been added to the text.*

**Reviewer's comment:**
  Line 390: I don't understand how this result is consistent with Figures 7 and 8, which have nothing to do with the bottom altitude.

  **Response:**
  *We wanted to say that the denser PMCs, which, as Fig. 8 shows, are closer to the poles, are located at lower mean altitudes, as shown in Fig. 7.*
  *We have clarified this in the revised version: "This is consistent with the behaviour shown in Figs. 7 and 8 where the denser layers are usually found near the poles and at lower mean altitudes."*

- **Reviewer's comment:**
  Line 460: This anti-correlation doesn't seem apparent to me.

  **Response:**
  *We now clearly state that the anti-correlation is generally good only in the NH but not in the SH. As mentioned above, we have re-written Section 9.*

- **Reviewer's comment:**
  Line 513: Didn't you show this in Figure 1 and 3?

  **Response:**
  *Not really in Figs. 1 because it shows some examples which might be or not a general behavior. About Fig. 3, yes, it shows the inter-hemispheric differences qualitatively although*

*not quantitatively. In any case, we are recapping the major results in the Conclusions' section and find it useful mentioning this again.*

- **Reviewer's comment:**
  Minor Changes:
  All the minor changes have been included. Thank you very much.

  FIgures:

[Figure]

**Figure 5.** Comparison of the ice mass density of MIPAS MUA modes of measurements (see Table 1) with SOFIE v1.3 L2 data for the 2008 to 2011 period and the mean of the four years in the NH. The solid lines show the mean profiles, SOFIE in black and MIPAS in red. The shaded areas are the standard deviations divided by the square root of the number of profiles. The means of the IWC are also shown.

*Fig. 1. New Fig. 5.*

[Figure]

*Fig. 2. Correlation between IWC and the altitude of the lower branch of the frost point temperature contour for the data taken in the NLC (black '+') and MUA (red diamonds) in the NH.*

[Figure]

*Fig. 3. As Fig. 2 but for the SH.*